# Non-Ceruloplasmin Copper Identifies a Subtype of Alzheimer’s Disease (CuAD): Characterization of the Cognitive Profile and Case of a CuAD Patient Carrying an *RGS7* Stop-Loss Variant

**DOI:** 10.3390/ijms24076377

**Published:** 2023-03-28

**Authors:** Rosanna Squitti, Claudio Catalli, Laura Gigante, Massimo Marianetti, Mattia Rosari, Stefania Mariani, Serena Bucossi, Gioia Mastromoro, Mariacarla Ventriglia, Ilaria Simonelli, Vincenzo Tondolo, Parminder Singh, Ashok Kumar, Amit Pal, Mauro Rongioletti

**Affiliations:** 1Department of Laboratory Science, Research and Development Division, Fatebenefratelli Isola Tiberina—Gemelli Isola, 00186 Rome, Italy; 2Osakidetza Basque Health Service, Department of Genetics, Cruces University Hospital, 48903 Barakaldo, Spain; 3Neuromuscular Disorders Research Group, Biocruces Bizkaia Health Research Institute, 48903 Barakaldo, Spain; 4Eurofins Genoma Group, Molecular Genetics Laboratory, 00138 Rome, Italy; 5Experimental Alzheimer Center, Fatebenefratelli Roman Province, 00189 Rome, Italy; 6Digestive and Colorectal Surgery, Fatebenefratelli Isola Tiberina—Gemelli Isola, 00186 Rome, Italy; 7Digestive Surgery Unit, Fondazione Policlinico Universitario A. Gemelli IRCCS, 00168 Rome, Italy; 8Centre for Systems Biology and Bioinformatics, Panjab University, Chandigarh 160025, India; 9Department of Biochemistry, All India Institute of Medical Sciences (AIIMS), Kalyani 741245, India

**Keywords:** Alzheimer’s disease, ceruloplasmin, copper, non-bound ceruloplasmin copper, metals, dementia, executive functions, dopamine, Iron, G-protein, *RGS7*, stop-loss mutation

## Abstract

Alzheimer’s disease (AD) is a type of dementia whose cause is incompletely defined. Copper (Cu) involvement in AD etiology was confirmed by a meta-analysis on about 6000 participants, showing that Cu levels were decreased in AD brain specimens, while Cu and non-bound ceruloplasmin Cu (non-Cp Cu) levels were increased in serum/plasma samples. Non-Cp Cu was advocated as a stratification add-on biomarker of a Cu subtype of AD (CuAD subtype). To further circumstantiate this concept, we evaluated non-Cp Cu reliability in classifying subtypes of AD based on the characterization of the cognitive profile. The stratification of the AD patients into normal AD (non-Cp Cu ≤ 1.6 µmol/L) and CuAD (non-Cp Cu > 1.6 µmol/L) showed a significant difference in executive function outcomes, even though patients did not differ in disease duration and severity. Among the Cu-AD patients, a 76-year-old woman showed significantly abnormal levels in the Cu panel and underwent whole exome sequencing. The CuAD patient was detected with possessing the homozygous (c.1486T > C; p.(Ter496Argext*19) stop-loss variant in the *RGS7* gene (MIM*602517), which encodes for Regulator of G Protein Signaling 7. Non-Cp Cu as an add-on test in the AD diagnostic pathway can provide relevant information about the underlying pathological processes in subtypes of AD and suggest specific therapeutic options.

## 1. Introduction

Alzheimer’s disease (AD) is the most common neurodegenerative condition associated with dementia in the elderly. Hallmarks of AD are extracellular aggregates of amyloid-β (Aβ), and intracellular neurofibrillary tangles made of hyperphosphorylated tau (p-Tau) [1,2]). Recent programs aiming at the development of non-invasive AD biomarkers reached important achievements by employing plasma Aβ 40-42 (Neurology 3-Plex A-performed with SIMOA technology [3]), plasma phosphorylated tau 217 and tau181, and the Lumipulse G β-Amyloid Ratio (42/40) [4]. Furthermore, the Clarity study, a confirmatory phase III clinical trial with Lecanemab, met the primary endpoint [5], showing a highly statistically significant reduction in the clinical decline in a global clinical study of 1795 participants with early AD, giving patients and their families a concrete hope for a cure. Despite these brilliant achievements, it should be considered that treatment with Lecanemab has reduced clinical decline on the global cognitive and functional scale Clinical Dementia Rating scale Sum of Boxes by only 27% [5]. This suggests that the remaining disease burden may have no disease-modifying therapeutic options. On this basis, tests aiming at improving diagnostic accuracy and identifying additional disease-associated metabolic sub-pathways, namely add-on tests, deserve consideration. An add-on test is a new test that is performed after the standard diagnostic pathway has been completed. It may be used to classify patients into disease subtypes and to tailor additional/alternative therapeutic procedures on ‘precision medicine’ (diet, supplement, drug re-purposing), and on ‘biomarker science’ [6,7,8].

Various stratification methodologies have been tried to establish subgroups of AD patients and several hypotheses have been proposed. A major emerging concept is referred to as the “Theory of metal imbalance in AD” (review in [8]). This theory is based on the evidence that a subpopulation of AD patients shows abnormal values of certain copper (Cu) biomarkers. This subtype of AD patients has been named the CuAD subtype (reviewed in [8]). The link between AD pathogenesis and Cu dysmetabolism has been addressed in a sizable number of studies recently reviewed in [8,9]. Patients with CuAD primarily exhibit higher than normal serum non-bound ceruloplasmin Cu (non-Cp Cu) (normal reference range 0.05–1.6 µmol/L [10,11,12]). Non-Cp Cu values higher than 1.6 µmol/L can be found in AD patients belonging to the CuAD subset, while values higher than 2.31 µmol/L most likely pertain to Wilson’s disease (WD) patients [10,11,12,13]. Non-Cp Cu (also known as ‘free Cu’) is the fraction of Cu in serum/plasma that does not bind to ceruloplasmin, the main protein that carries Cu in the blood (reviewed in [14]). Due to the redox active property of Cu, it is generally considered toxic above the cut-off as it can cross the blood–brain barrier (BBB) and damage the brain parenchyma, as exemplified in WD [10,11,12].

The “Theory of metal imbalance in AD” is well supported by a wealth of clinical and meta-analysis studies [8,15] showing cerebral Cu deficiency and increased serum non-Cp Cu as signs of Cu dyshomeostasis in AD (recently reviewed in [8]). Importantly, stratification into the CuAD subtype is possible as serum non-Cp Cu levels can be assessed in living patients, contrary to AD brain deficiencies.

Cu, an essential micronutrient, is a vital component of numerous metalloproteins/enzymes that plays crucial roles in various biological systems. Cu is imported into the cell by CTR1 [16]. In the cytoplasm, the P-type ATPase (ATPase) 7A and ATP7B pump Cu^2+^ into the trans-Golgi network or out of the cell membrane via the vesicular compartment. The endocytosis mechanisms of G-protein-coupled receptors (GPCRs) parallel Cu uptake based on CTR1 [17,18,19]. GPCR-mediated Cu endocytosis has been described for gonadotropin-releasing stimulating hormone (GnRH) [20], for neurokinin (NK) A and B [21], and for neuropeptide Y [20,21]. Cell signaling mediated by GPCRs can be terminated by several mechanisms, one of which involves hydrolysis of the Gα-bound GTP and reassembly of the inactive Gαβϒ trimer [22], facilitated by Regulators of G-Proteins signaling (RGS) proteins [23,24,25]. RGS proteins typically accelerate the intrinsic GTPase activity of the Gα subunit, resulting in an inhibition of downstream G protein signaling pathways [26,27].

In this study, we propose a new pilot diagnostic pathway to improve the diagnostic accuracy of AD, based on non-Cp Cu as a stratification add-on biomarker. After applying the standard AD diagnostic pathway [28], patients underwent a Cu panel screening and an extensive neuropsychological battery including Mini-Mental State Examination (MMSE) [29], Montreal Cognitive Assessment (MOCA) [30], Frontal Assessment Battery (FAB) [31], Clock-Drawing Test (CDT) [32], and Neuropsychiatric Inventory (NPI) [33]. We then evaluated non-Cp Cu reliability in classifying subtypes of AD based on the characterization of the cognitive profile. Furthermore, patients with abnormal values in at least two biomarkers of the Cu panel underwent whole exome sequencing (WES) analysis to screen for potential causative gene variants. During the trial of this pilot clinical diagnostic path, we identified a patient with an *RGS7* stop-loss gene variant associated with CuAD. Results also support the use of non-Cp Cu assessment as a stratification add-on test for the Cu-AD subtype, based on the characterization of the cognitive profile.

## 2. Results

During the trial period of our pilot diagnostic pathway, 30 patients were referred to our site. Twenty-eight received the diagnosis of AD (IGW2-criteria [28]). We first compared AD patients and healthy controls (CTRL) for the biomarkers of the Cu panel. We then used the non-Cp Cu biomarker as an add-on test to stratify the AD patients into two subgroups on the basis of a 1.6 μmol/L cut-off: we then compared the two AD subgroups for the outcomes of the extensive neuropsychological battery. One of the patients of the CuAD subgroup, a 76-year-old woman, showed significantly abnormal levels in more than two Cu biomarkers; she then underwent WES and was found to have an *RGS7* stop-loss gene variant.

### 2.1. Alzheimer’s Disease and Healthy Controls Comparison for Cu Panel

The AD and CTRL groups differed in sex and age, with the healthy group being younger than the AD group and 76% of the CTRL group being women (Table 1). For this reason, the analyses were adjusted to account for the potential confounding effect of age and sex. Serum Cu concentration was different between the two groups (Welch’s *t*-test), being higher in AD patients than in CTRL. The two groups differed significantly in non-Cp Cu and Cu:Cp ratio values, which were higher in AD than in CTRL; ceruloplasmin biomarkers did not differ between the two groups (Figure 1 and Table 2).

### 2.2. AD Subgroups Stratification on the Basis of the Add-On Non-Cp Cu Test

The AD population was stratified on the basis of the non-Cp Cu cut-off of 1.6 µmol into two subtypes (Table 3). Among the CuAD patients, one had severe Cu abnormalities, thus was studied apart as a case report (see Section 2.3).

Subjects in the CuAD subgroup had higher values of Cu, ceruloplasmin, ceruloplasmin activity, and specific ceruloplasmin activity than patients in the AD subgroup. In addition, non-Cp Cu was about three times higher in the CuAD group than in subjects in the AD subgroup (1.14 ± 0.49 in AD vs. 3.29 ± 0.94 CuAD, *p* < 0.001). The two AD subgroups had the same years of disease duration and the same degree of severity (as indicated by the MMSE score, which did not differ between the two groups), but differed in neuropsychology test scores (Table 4), with FAB and CDT scores being lower in the CuAD subgroup (Table 4 and Figure 2). After the Benjamini & Hochberg’s [34] *p*-value adjustment for multiple testing (B-H adjusted), only the FAB difference remained significant (Table 4). We calculated Cohen’s d [35] to provide a standardized measure of the size of the difference between groups. Cohen’s d demonstrated a large difference between the normal AD and the CuAD for MMSE (d = 0.84, 95%CI: 0.01; 1.68), FBA (d = 1.24, 95%CI: 0.37; 2.11), and CDT (d = 0.99, 95%CI: 0.15; 1.85), while it was medium for MOCA (d = 0.59, 95%CI: −0.23; 1.41) and small for NPI (d = 0.26, 95%CI: −1.06; 0.55). Results were confirmed by repeated measures ANOVA. The interaction of the patient group (between-subject: CuAD vs. AD) and neuropsychological tests (within-subject variable) was significant (*p* = 0.021). Specifically, post-comparisons showed that CuAD patients performed significantly worse on FAB (B-H adjusted *p* = 0.006) as well as on CDT (B-H adjusted *p* = 0.024), and borderline on MMSE (B-H adjusted *p* = 0.055).

We performed an exploratory correlation study between neuropsychological test results and Cu panel biomarkers. Analyses were performed on all AD patients without considering subgroup stratification (Table 5). Some tests of the neuropsychological battery had trends of correlation with Cu biomarkers, namely non-Cp Cu and ceruloplasmin-specific activity (Table 5). After applying Benjamini & Hochberg’s [34] correction to control for the false discovery rate, the only worse scores on the FAB scale correlated with abnormal values of all Cu biomarkers (Table 5).

### 2.3. Case Report of One Patient in the CuAD Subgroup

The patient was referred to our specialty memory clinic at the Experimental Alzheimer Center, Fatebenefratelli Roman Province, Rome, Italy at the age of 76 with a diagnosis of AD according to IWG-2 criteria [28] and a 14-year history of the disease.

She was positive for AD biomarkers in the CSF, namely decreased levels of Aβ40/42 and increased levels of P-tau or T-tau and had a positive FDG-PET (Figure 3), compatible with the diagnosis.

Her past medical history revealed no relevant pathological conditions. She had no surgical history. She had no history of smoking, alcohol, or drug use. There was no first-degree family history of presenile dementia. Laboratory screening for markers of liver functions, magnesium, and phosphorus was normal, while iron metabolism markers and lithium were slightly below the normal reference range (Table 6).

After completing the AD diagnostic pathway, the patient underwent a Cu panel screening and an extensive neuropsychological battery as per the add-on test Cu diagnostic pathway. The Cu study revealed more than two Cu biomarker abnormalities, namely non-Cu Cp (higher than 1.6 μmol/L) [11,36], Cu (lower than 11 μmol/L [37]), and ceruloplasmin activity that was higher than 140 IU [38] (Table 6). On this basis, she underwent WES analysis to screen for potential causative mutations.

### 2.4. Analysis and Identification of a Candidate Variant in the CuAD Patient

The initial Next Generation Sequencing panel of 74 genes (see method section) did not identify any pathogenic, likely pathogenic, or unknown significance variants. No common functional polymorphisms were found in the *ATP7B* gene.

Whole Exome Sequencing (WES) allowed the identification of 49 candidate single-nucleotide variants (SNVs), with further refinement according to in silico predictors, detailed population database, and intolerance prediction to two top-candidate SNVs: the nonsynonymous heterozygous c.368G>A; p.(Arg123Gln) NM_001005361.3; rs777834215 *DNM2* (Dynamin 2) variant, of unknown significance (American College of Medical Genetics and Genomics, PM2 moderate, PP3 moderate) and the predicted stop-loss homozygous c.1486T>C; p.(Ter496Argext*19) NM_001364886.1; rs1294663627 variant in *RGS7* (Regulator of G protein signaling 7), classified as probably pathogenic (ACMG criteria, PM4 very strong, PM2 supporting).

The *DNM2* variant shows an allelic frequency of 2:113.728 alleles for the European population according to GnomAD exomes, which corresponds to around one carrier in 29,000, moderately compatible with AD disease with late-onset, and highly compatible with AR disease frequency carrier. All in silico tools predict a possible damaging effect. *DMN2* is associated with myotubular myopathy type 1 (MIM #160150) with an autosomal dominant trait of inheritance, Charcot-Marie-Tooth disease, axonal type 2M (CMT2M), Charcot-Marie-Tooth disease, dominant intermediate B (CMTDI1) (MIM #606482), and with Lethal congenital contracture syndrome type 5 (MIM #615368) following an autosomal recessive fashion. The patient showed no neuromuscular or polyneuropathy clinical signs.

The *RGS7* variant presents an allelic frequency of 4:68.002 alleles for the European population according to GnomAD exomes, with no homozygous cases, which corresponds to around one carrier in 9,000, doubly compatible with AD disease with late-onset and highly compatible with AR disease frequency carrier. To date, no MIM phenotype has been linked to *RGS7* pathogenic variants (Figure 4). The combined Annotation Dependent Depletion (CADD) score was 22.1, predicting this variant to be in the top 1% for deleteriousness [39].

### 2.5. In Silico Analysis of RGS7 Stop-Loss Mutation

Change at the 496th position in a wild-type sequence results in the loss of a stop codon and changes it into arginine (Arg) and extends the stretch of the protein sequence by 19 amino acids. In order to find similarly related sequences, Protein Blast was utilized. These sequences were then used for multiple sequence alignment, showing that 14 amino acids in the extended protein sequence were conserved in most of the sequences shown in Figure 5.

Only four of these 19 amino acids were in a helical and others were in a coil, according to secondary structure prediction. The dynamics of the protein were considerably different between mutant and wildtype proteins according to the B-factor study shown in Figure 6, which may result in a protein loss of or gain of function.

The Extended protein structure demonstrated the difference between the wild-type and the altered *RGS7* protein (Figure 7a–c).

Secondary structure prediction of extended protein sequence exhibited most of the amino acids were in the coil (C) structure, and only four residues were present in the helix (H) structure (Figure 7d). Solvent accessibility ranged between 0–9. Values range from 0 (buried residue) to 9 (highly exposed residue) (Figure 7e).

## 3. Discussion

The main result of this study is that AD patients displayed neuropsychological heterogeneity after stratification into two subgroups based on the non-Cp Cu add-on test, demonstrating our working hypothesis. Another result is the identification of an *RGS7* stop-loss variant in a CuAD patient with severely abnormal Cu, supporting a potential role of the *RGS7* protein in regulating Cu trafficking. Furthermore, current data suggest an association of the *RGS7* stop-loss variant with CuAD. However, the data presented are merely descriptive and further experimental evidence is needed to claim a causative connection between the *RGS7* stop-loss variant and AD onset or progression, as well as the role of *RGS7* as a protein involved in regulating Cu trafficking.

### 3.1. Comparison of the AD Subtypes

CuAD subtype patients scored lower on the FAB test than AD subjects with normal non-Cp Cu, despite the fact that the two subgroups did not differ in disease duration and severity of cognitive decline, as evidenced by the MMSE score. CDT was also lower in CuAD than in typical AD, but it did not reach the statistical threshold after the B-H adjustment, likely because of the small size of the sample. Furthermore, after applying Benjamini & Hochberg’s [33] correction, we noted that worse scores on the FAB scale correlated with abnormal values of all Cu biomarkers. The FAB test and the CDT specifically assess executive functions, i.e., the set of mental processes aimed at developing adaptive cognitive-behavioral patterns in response to novel and challenging environmental conditions. The prefrontal cortex, located in the frontal lobe, plays a key role in executive functions. Dopamine and noradrenalin are catecholamines that play a crucial role in the cortical-subcortical circuitry involved in executive functions. Cu is a component of metallo-proteins and a cofactor of enzymes (dopamine β-hydroxylase, monoamine oxidase) involved in catecholamine balance [41]. Thus, it is reasonable that the Cu imbalance observed in the CuAD subtype could result in an alteration of the catecholamine balance and explain the executive function deficits exhibited by these patients. As a matter of fact, catecholamine balance is altered in other disorders of Cu metabolism, primally in WD [41].

### 3.2. Case Report: DNM2 and RGS7 Mutation and Alzheimer’s Disease

Extensive analysis of WES data allowed the identification of a small number of candidate genes.

*DNM2* pathogenic variants cause a spectrum of neuromuscular diseases but no involvement in dementia or altered Cu metabolism has been documented.

To the best of our knowledge, *RGS7* is a gene with no phenotype associated to date.

Expression of *RGS7* is especially abundant in the frontal cortex and overall, in the brain, where other RGS proteins are also temporally and spatially expressed [42]. It has been linked to a number of functions, such as synaptic plasticity, memory, learning, and maintenance of dopaminergic neurons, and is thus involved in addiction and psychiatric diseases [43,44,45].

This protein can also form a complex with 14-3-3 proteins, which has been linked to different neurodegenerative disorders including Parkinson’s and Lewy body [46,47,48].

### 3.3. Detection of RGS7 Variants Highlights the Relevance of Alternative Pathways of Intracellular Cu Trafficking in Alzheimer’s Disease

We have described in detail elsewhere [8] the main biological processes that connect Cu imbalance with major degenerative processes related to the Aβ and tau proteins in AD. The finding of *RGS7* involvement in CuAD adds new insight into a less explored pathway that could help elucidate the role of non-Cp Cu in this disease.

The current results support growing evidence on the role of the endocytosis mechanisms of GPCRs in cellular Cu uptake and trafficking [16,19]. The transport machinery of GPCRs has been proposed to control a clathrin-dependent pathway in which NK A and B, and the neuropeptide Y, pair with Cu and enter the cell by pinocytosis. This Cu/peptide uptake has been described primarily in nerve cells and in the hippocampus [21]. NK A and B and GNrH are low molecular weight compounds that bind Cu and travel into the blood circulation as part of the non-Cp Cu serum component [20,21]. Furthermore, it has been shown that NK B can limit Cu uptake into astrocytes [21]. A new hypothesis poses that specialized astrocytes producing fatty acid-binding protein 7 (FABP7) might be related to AD pathogenesis [49]. FABP7+ astrocytes have been proposed to support adult neurogenesis in the dentate gyrus of the hippocampus. It is known that astrocytes are the main controller of Cu homeostasis in the brain (review in [50]). Abnormal non-Cp Cu levels may threaten Cu brain homeostasis. We do not have causal evidence demonstrating a direct link between the *RGS7* loss-stop variant and the onset of AD. Nevertheless, we cannot rule out that the high non-Cp Cu serum concentration (4 µmol/L, Table 6) of the patient carrying the *RGS7* loss-stop variant, along with other *RGS7* altered mechanisms, could have had effects in accelerating AD pathogenesis. In line with previous evidence [20,21], FABP7+ astrocytes might control the Cu/ NK A and B and other neuropeptides up-take through endocytosis, which is dependent on GPCRs and controlled by the *RGS7* protein. Theoretically, it could be biologically plausible that FABP7+ astrocytes may be involved in restraining excess labile Cu in the brain. We are still far from defining *RGS7* as a new gene causally linked to Cu metabolism and AD. Nonetheless, our results partially corroborate the new FABP7+ astrocytes’ hypothesis [49]: non-Cp Cu serum expansion associated with the *RGS7* stop-loss variant might decrease the ability of FABP7+ astrocytes to clear labile brain Cu, and FABP7+ astrocytes might no longer protect neurogenesis in the hippocampus, accelerating the disease cascade [49,50].

### 3.4. The Presence of the RGS7 Stop-Loss Variant Highlights the Relevance of Alternative Pathways of Intracellular Fe Trafficking in Alzheimer’s Disease

Another finding of the current study is that the AD patient carrying the *RGS7* stop-loss variant has lower levels of circulating markers of iron metabolism. Iron, transferrin, TIBC, and transferrin saturation are low, while ceruloplasmin activity measured by the ortho-dianisidine assay (ceruloplasmin-referred oxidative activity) is strongly increased. The serum markers of Cu and Fe in this patient are very peculiar and both could be explained by the dysregulation of metal transport controlled by the clathrin pathway, which is dependent on GPCRs and controlled by the *RGS7* protein. Further research is needed to explore RGS7′s involvement in metal serum concentration.

A limitation of our study regards the size of the sample and possible sampling bias related to mild cases, which are underrepresented. The results of our pilot study need to be confirmed in other populations and especially in early cases. In addition, a study on the frequency of the *RGS7* stop-loss variant in a larger population of patients with CuAD is in progress.

## 4. Materials and Methods

Non-Cp Cu has been advocated as a stratification add-on biomarker of the CuAD subtype (review in [13]). To further circumstantiate this concept, we performed a pilot clinical study focused on improving diagnostic accuracy for use in precision medicine applied to AD. The primary goal of precision medicine is to treat patients with the appropriate drugs based on their individual biological and genetic makeup. To this end, we have developed a pilot diagnostic pathway for future therapeutic programs based on anti-Cu drugs. In brief, our pilot diagnostic pathway can be summarized, as follows:
Application of the standard diagnostic pathway for AD according to the IGW2 criteria [28];Apply the stratification add-on biomarker to classify the AD population into two subgroups on the basis of the established non-Cp Cu cut-off (1.6 µmol/L) [13];Search for genetic variants that might explain the disease-associated metabolic sub-pathways, i.e., searching for mutations that may explain the altered Cu metabolism.

Furthermore, in order to collect additional evidence on the CuAD subtype, we applied one of the four strategies devoted to this scope and detailed elsewhere [13,51]. In brief, after classifying AD patients into subgroups based on the non-Cp Cu cut-off, we compared the subgroups for the neuropsychological testing outcomes to find heterogeneity, as heterogeneity speaks in favor of disease subtypes [51].

### 4.1. Subjects

A total of 41 healthy control subjects (CTRL) and 30 consecutive patients with dementia were recruited at the Experimental Alzheimer Center, Fatebenefratelli Roman Province, Rome, Italy. Twenty-eight of them received a final diagnosis of AD according to International Working Group (IWG) 2 Criteria for AD (IWG-2 criteria) [28], with gradual progressive change in memory function for >6 months, objective evidence of hippocampal amnestic syndrome, a marked reduction in cerebrospinal (CSF) Aβ42 [52,53] and in the Aβ42/Aβ40 ratio [28], increased levels of P-tau or T-tau in the CSF, or a positive amyloid positron emission tomography (PET) or fluorodeoxyglucose (FDG)-PET, and MMSE ≤ 25, and were enrolled. Two patients did not meet the IGW2 AD criteria and were excluded. Healthy controls were individuals with no sign of neurological disorders and with normal cognitive function, selected mainly among spouses. Exclusion criteria were conditions affecting Cu metabolism, evaluated on the basis of past medical history. Participants underwent blood sampling after an overnight fast, and analyses of an extensive panel of Cu metabolism biomarkers. Patients underwent an extensive neuropsychological battery MMSE [29], MOCA [30], FAB [31], CDT [32], and NPI [33]. The study protocol was approved by the local ethics committee and subjects provided written informed consent.

### 4.2. Biochemical Analyses

The panel of Cu metabolism biomarkers is made of: Ceruloplasmin concentration (Cp): Immunoturbidimetric assay (Futura System S.r.l.); Cu concentration in serum measured by Abe [54] Colorimetric assay (Randox) and confirmed in 30% of the samples by Atomic absorption Spectrometry (A Analyst 600 Perkin Elmer); non-Cp Cu Calculated: Cu–(0.472*iCp) as proposed by Walshe [11]; Copper:ceruloplasmin ratio (Cu:Cp) is an additional Cu index that serves as an useful internal quality control to verify ceruloplasmin calibration, and offers information about the actual stoichiometry between Cu and ceruloplasmin in the specimens: a Cu:Cp value of 6.6 is the theoretical optimal ratio for healthy subjects [55]. Cu:Cp calculation: Cu/Cp [MW of Cp (132KDa)] as proposed by Twomey [56]. Non-Cp Cu (exchangeable Cu) was measured by Ultrafiltration +AAS600 Perkin Elmer measure of Cu (adapted from [36]). The percentage of transferrin saturation was calculated as follows: the value of serum iron divided by the total iron binding capacity [TIBC = transferrin (mg/dL) × 1.25] × 100.

### 4.3. Statistical Analysis

The normality of the distribution of variables was assessed by the Kolmogorov-Smirnov test (*p* > 0.05). Association between the two groups and dichotomous variables were assessed by Chi-square or, when appropriate, by Fisher’s exact test. Differences between the groups in continuous variables were assessed by parametric *t*-test or, when the two samples had different variances, by Welch’s *t*-test.

To account for multiple comparisons in the analysis of cognitive abilities between AD and CuAD, the Benjamini & Hochberg’s [34] correction was applied to the *p* values. A *p*-value < 0.05 was considered statistically significant.

To compare the difference found among the two groups (AD, CuAD), Cohen’s d and the corresponding 95 Confidence Interval (95% CI) were calculated. The interpretation was based on benchmarks suggested by Cohen [35]: d = 0.2 indicates a small effect size, d = 0.5 medium, and d = 0.8 large. In addition, the repeated measures ANOVA model was performed, considering the type of neuropsychological tests as a within-subject factor and the groups as a between-subject factor. Since the neuropsychological tests had different ranges of variations, for each test a normalization was applied, as follows: (observed value-Scale Minimum value)/(Scale Maximum value-Scale Minimum value).

A Pearson’s correlation coefficient was calculated to evaluate the association between the neuropsychological test battery vs. each biomarker of the Cu panel (Cu, non-Cp Cu, ceruloplasmin, ceruloplasmin activity, and ceruloplasmin-specific activity). To control for the false discovery rate (FDR), Benjamini & Hochberg’s correction of the *p*-value was applied.

Statistical analyses were performed with IBM SPSS statistic version 16.0 and version 9.3.1 of the GraphPad Prism program for graphs.

The main purpose of our pilot study was to identify two subtypes of AD patients using a non-Cp Cu biomarker as a stratification add-on test (cut-off 1.6 µmol/L) after the application of the mainstream diagnostic pathways according to the IGW-2 criteria [28]. Our working hypothesis was that AD patients display neuropsychological heterogeneity after their classification into two subgroups based on the non-Cp Cu add-on test. At the end of recruitment, 30 consecutive patients have been enrolled: 2 patients were excluded since they did not meet the IGW2 criteria, an additional one (a 76-year-old CuAD patient) was not included in the statistical analysis since she was studied separately as a case report, thus 27 patients entered the classification study.

### 4.4. Case Report and Genetic Analysis

Genomic DNA was yielded by automatic DNA extraction employing Qiagen–Qiasymphony. Library preparation was performed with the Illumina^®^ DNA Prep with Enrichment (Illumina, San Diego, CA, USA) protocol using xGen™ Exome Research Panel v2 (IDT, Integrated DNA Technologies, Inc., Coralville, Iowa, USA) to enrich the sample for about 19,433 genes (34 Mb target region). Massively parallel sequencing was carried out on NextSeq 550 (Illumina, San Diego, CA, USA) with paired End method (2 × 101 cycles). BaseSpace software (Illumina San Diego, CA, USA) and BaseSpace Variant Interpreter (Illumina©, San Diego, CA) allowed us to generate bam, bam.bai and vcf files. GRCh37 (hg19) was used as the reference sequence for mapping and variant calling.

Databases accessed for analysis and variant annotation include the Online Mendelian Inheritance in Man (OMIM), Human Gene Mutation Database, ClinVar, Single Nucleotide Polymorphism database (dbSNP), and Genome Aggregation Database (gnomAD), with a minor allele frequency cut-off of 1%.

The pathogenicity of variants was predicted using the following in silico prediction algorithms: VarSome and Franklin variant interpretation (Genoox).

We used VarAft to annotate VCF from WES [57].

The targeted analysis included a list of 74 genes selected using HPOs HP:0002511 (Alzheimer’s disease), HP:0000726 (Dementia), HP:0010837 (Frontotemporal Dementia), HP:0010838 (High non-ceruloplasmin serum Cu), HP:0010837 (Decreased serum ceruloplasmin).

For WES analysis, we included all exons and surrounding 10 intronic bases, with a minimum vertical coverage of 10×, discharging all calls without PASS quality and with an allele fraction (AF) < 0.3. For a dominant mode of inheritance, we selected all variants with MAF < 0.00001 assuming possible presence in GnomAD due to late onset of disease, while for an autosomal recessive mode of inheritance, we selected all variants with MAF < 0.005 with no more than 1 homozygous present in GnomAD database. Variants were classified according to ACMG criteria, and we kept only P, PP, or VUS variants. Gene prioritization was made using Gene Ontology, GTEx, and REACTOME according to their expression, localization and function, and also manually curated according to the published literature. In silico prediction was realized using the UMD predictor, Human Splicing Finder, SIFT, Polyphen 2 HumDiv, Polyphen 2 HumVar, Mutation Taster, Mutation Assessor, Provean, M-CAP, LRT, CADD, and DANN. For genes with unknown modes of inheritance, in the case of heterozygous mutations, we used RVIS score and LoFTool to predict possible intolerance to non-synonymous variants and loss-of-function.

### 4.5. In Silico Analysis of RGS7 Stop-Loss Mutation

Data collection: The nucleotide sequence and protein sequence of *RGS7* was downloaded from NCBI [58]. For the mutant sequence, T is replaced with C (T > C) at position 1432 (Ter496Argext*19) in the wildtype sequence and translated into the protein sequence, and Open Reading Frame (ORF) analysis was carried out using the expasy tool [59].

BLAST Analysis: The Basic Local Alignment Search Tool (BLAST) identifies areas where sequences are locally similar. It can be used to infer functional and evolutionary relationships between sequences. It is used to find closely similar sequences to *RGS7* [58].

Multiple Sequence Alignment (MSA) analysis: The alignment of three or more biological sequences (protein or nucleic acid) of comparable length is known as multiple sequence alignment. The results allow for the study of the evolutionary links between the sequences and the inference of homology. Clustal was used for the analysis of MSA [60].

B-factor and secondary structure analysis: The B-factors of protein crystal structures reflect the fluctuation of atoms about their average positions and provide important information about protein dynamics [61]. Secondary server structure, solvent accessibility prediction, and B-factor analysis were done by using protocols as described in [62]

Three-dimensional structure prediction: Robbeta was used to predict 3D structures. To predict the three-dimensional protein structure, this server uses homologous sequences or homology modeling [63]. Using the Chimera tool, wild-type and mutant structures were superimposed [64].

## 5. Conclusions

For the same disease duration and severity, the AD and CuAD subgroups differed in terms of cognitive profile, which was worse specifically in executive functions in subjects with higher non-Cp Cu. This heterogeneity argues in favor of the existence of a CuAD subtype, confirming previous evidence (review in [13]). The CuAD subtype had worse performance in FAB, and borderline changes in CDT, likely related to catecholamine circuit alteration associated with Cu imbalance. A patient of the CuAD subtype with severely abnormal Cu biomarkers turned out to be a carrier of an *RGS7* stop-loss variant that might explain her abnormal Cu serum profile, likely associated with a disturbed Cu uptake and cell distribution regulated by *RGS7*. It is known that non-Cp Cu crosses the BBB [10,11,12,13], so non-Cp Cu serum expansion, with levels higher than 1.6 µmol/L, may threaten Cu homeostasis in the brain. A new hypothesis poses that FABP7+ astrocytes in the dentate gyrus may participate in adult neurogenesis, and that might be related to AD pathogenesis [49]. Since astrocytes are the main controller of Cu homeostasis in the brain, abnormal non-Cp Cu levels may be associated with FABP7+ astrocytes in a disease-associated metabolic sub-pathway, eventually leading to AD. Overall, this study supports potential AD prevention strategies, such as a low-Cu diet and Zinc therapy, to be offered specifically only to individuals with CuAD/mild cognitive impairment (MCI). We are currently carrying out a Zinc therapy phase II clinical trial in CuMCI (Alzheimer’s Association Part the Cloud: Translational Research Funding for Alzheimer’s disease PTC-19-602325).

## Figures and Tables

**Figure 1 ijms-24-06377-f001:**
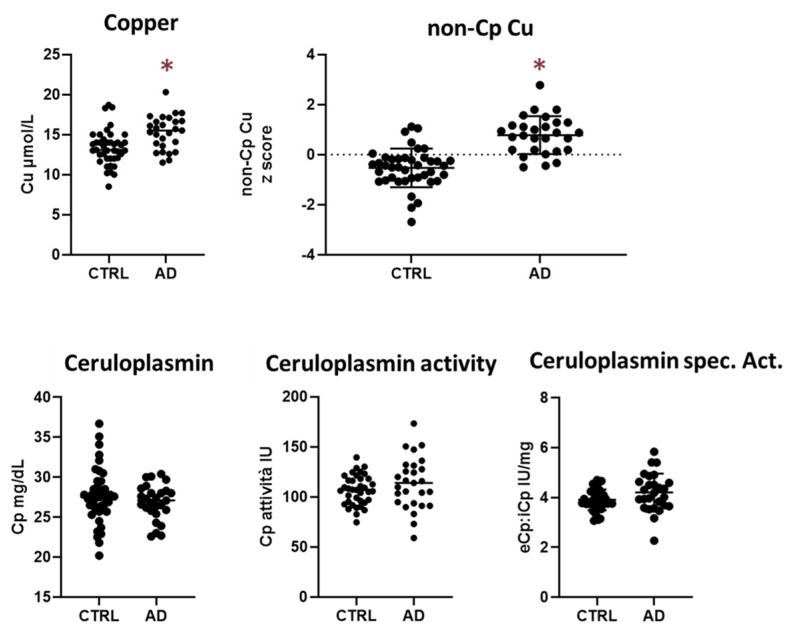
Comparison of the Cu panel between CTRL and AD subjects. Data are mean and standard deviation of Cu, non-Cp Cu, ceruloplasmin, ceruloplasmin, and ceruloplasmin-specific activity (* *p* < 0.05). Ceruloplasmin activity is additional quality control of ceruloplasmin measurement as extensively discussed elsewhere [11]. Ceruloplasmin-specific activity (eCp:iCp) is the enzymatic activity per mg of Cp concentration. Raw data are shown, not adjusted for subjects’ differences in sex and age.

**Figure 2 ijms-24-06377-f002:**
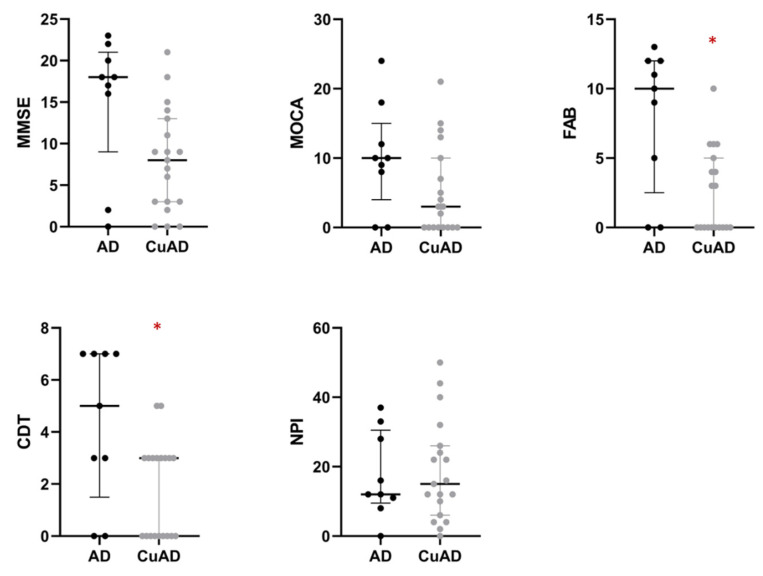
Comparison of AD and CuAD neuropsychological test battery. Data are mean and standard deviation. Raw data are shown, not adjusted for subjects’ differences in sex and age. AD: AD subjects with non-Cp Cu ≤ 1.6 μmol/L (black circles). CuAD: AD subjects with non-Cp Cu > 1.6 μmol/L (gray circles). MMSE: Mini-Mental State Examination. MOCA: Montreal Cognitive Assessment. FAB: Frontal Assessment Battery. CDT: Clock Drawing Test. NPI: Neuropsychiatric Inventory Test. * *p* < 0.05; after B-H adjustment only the difference in FAB was statistically significant (*p* = 0.040).

**Figure 3 ijms-24-06377-f003:**
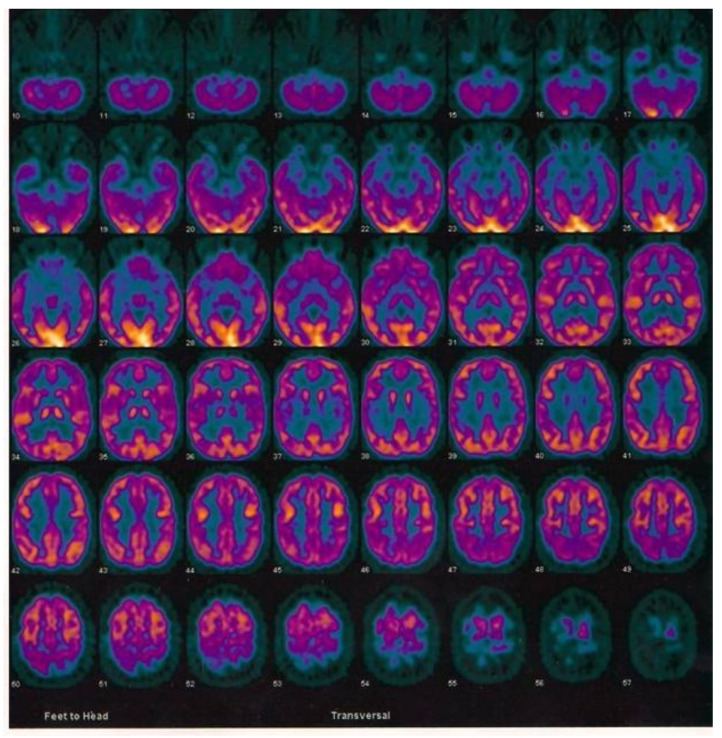
Results of fluorodeoxyglucose (FDG)- PET of the 76-year-old woman in the CuAD subgroup who was the carrier of the *RGS7* stop-loss mutation. Note the marked bilateral hypometabolism of different regions in the frontal and temporoparietal cortex.

**Figure 4 ijms-24-06377-f004:**
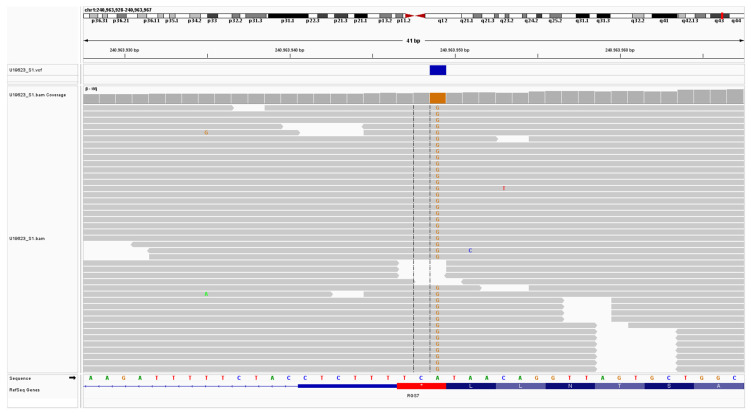
Integrative Genomics Viewer (IGV) [40] snapshot of the alignments showing the homozygote A to G change at position 1486 causing a stop loss at position 496 in the *RGS7* gene. Colorful letters refer to the nucleic acid at each position of the gene: (A—Adenine; G—guanine; C—cytosine; T—Thymine). Note:The variant of the RSG7 gene is reported in the manuscript as c.1486T>C; p.Ter496 Argext*19 according to 5′→3′ gene sequence as defined in HGVS nomenclature guidelines; the images provided show the complementary sequence of the gene.

**Figure 5 ijms-24-06377-f005:**
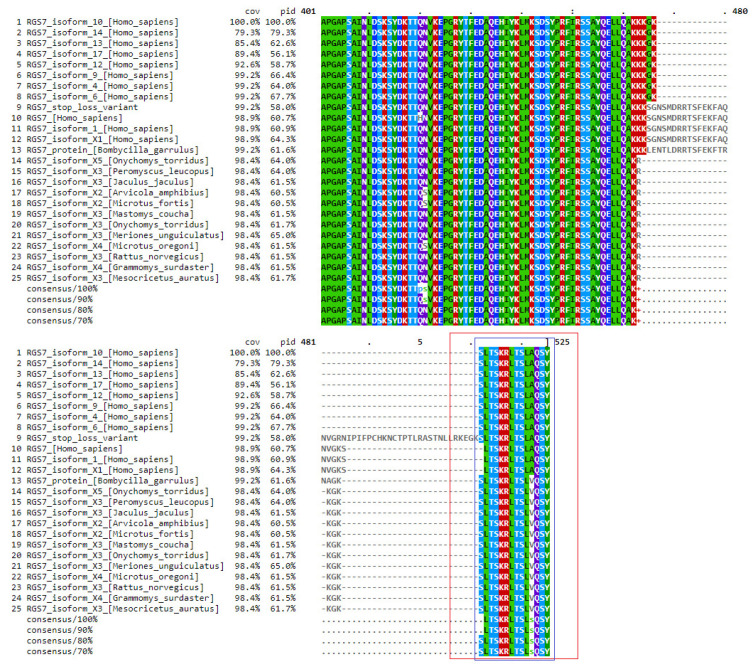
MSA shows that 14 amino acids in the extended protein sequence of *RGS7* were conserved in many organisms’ reference sequences. The red box shows the extended protein sequence of *RGS7* and the blue box shows the conserved 14 amino acids in the extended protein sequence of *RGS7*. The extended protein sequence is started from the 504th position of alignment and ends at the 522nd position of alignment (19 amino acids long), shown in the red box. The first five amino acid residues of extended protein sequence (R (Arginine), K (Lysine), E (Glutamic), G (Glycine), K (Lysine) are not conserved in any sequence but the last 14 amino acid residues are conserved in all of the sequences except for amino acid residue A (Alanine) at position 519 of the alignment which is conserved in Homo Sapiens but it is replaced by V (Valine) in the rest of the sequences. The protein sequence is conserved in vertebrates (Chondrichthyes, Reptiles, Aves, and mammals).

**Figure 6 ijms-24-06377-f006:**
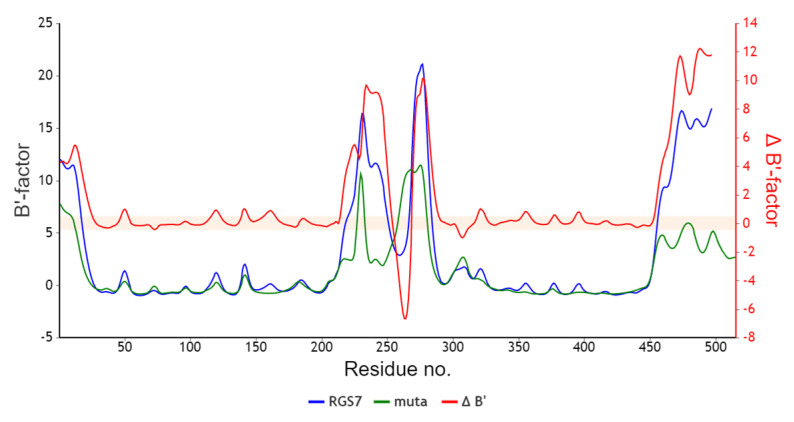
B-factor shows a significant difference between mutant and wild-type *RGS7* protein.

**Figure 7 ijms-24-06377-f007:**
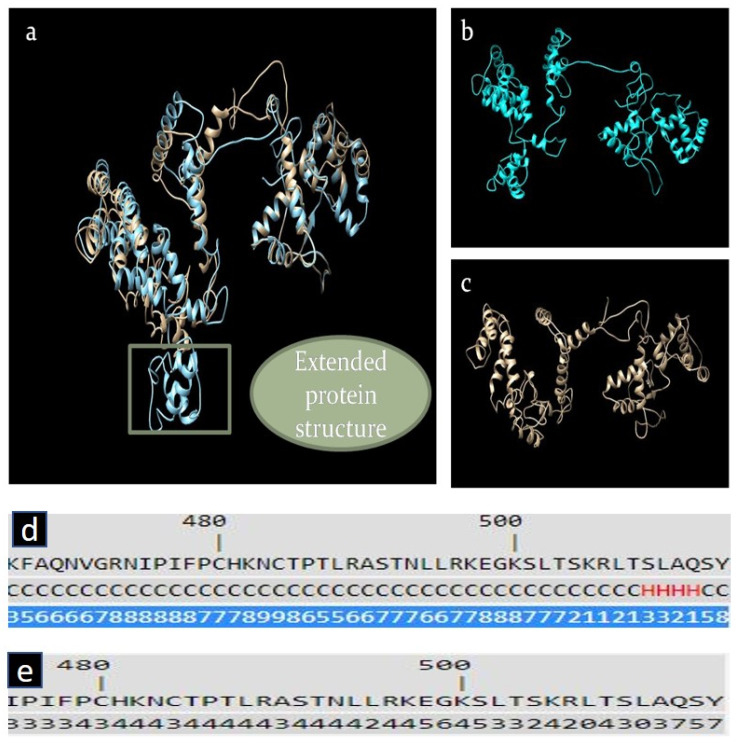
(**a**) Superimposed structure of wild-type and mutant *RGS7* proteins. (**b**) Mutant *RGS7* protein structure. (**c**) Wild-type *RGS7* protein structure. (**d**) Secondary structure prediction of extended protein sequence demonstrated that most of the amino acids were in a coil (C) structure, and only 4 residues were present in the helix (H) structure. (**e**) Solvent accessibility ranged between 0–9. Values range from 0 (buried residue) to 9 (highly exposed residue).

**Table 1 ijms-24-06377-t001:** Demographical characteristics of the study subjects.

	Heathy Controls (CTRL)	Alzheimer’s Disease (AD)	*p* Value
n	41	27	
Men, n (%)	10 (24%)	13 (48%)	
Women, n (%)	31 (76%)	14 (52%)	0.043
Age (±SD)	70.05 ± 8.8	77.78 ± 7.79	<0.001

SD, standard deviation.

**Table 2 ijms-24-06377-t002:** Comparison of the Cu panel between CTRL and AD subjects.

	Heathy Controls (CTRL)	Alzheimer’s Disease (AD)	*p* Value	Adjusted *p* Value ^b^
Cu, µmol/L	13.37 ± 2.18	15.27 ± 2.10	0.001	0.002
Cp, mg/dL	27.62 ± 3.45	26.87 ± 2.20	0.322	0.536
Cp-activity, IU	106.60 ± 14.86	113.60 ± 25.88	0.212	0.222
Cu:Cp ratio	6.4 ± 0.65	7.49 ± 0.60	<0.001	<0.001
non-Cp Cu, µmol/L	0.35 ± 1.34	2.58 ± 1.31	<0.001 ^a^	<0.001
eCp:iCp, IU/mg	3.91 ± 0.42	4.20 ± 0.76	0.076 ^a^	0.142

^a^ Welch’s *t*-test; ^b^ an ANCOVA model was run to assess the effect of diagnosis adjusted for sex and age; eCp:iCp: ceruloplasmin specific activity.

**Table 3 ijms-24-06377-t003:** Comparison of AD (non-Cp Cu < 1.6 µmol/L) and CuAD (non-Cp Cu ≥ 1.6 µmol/L).

AD Subtypes	AD	CuAD	*p* Value
n	9	18	
Men, n (%)	7 (78%)	6 (33%)	0.046 ^a^
Women, n (%)	2 (22%)	12 (67%)	
Age (±SD)	79.22 ± 8.33	77.06 ± 7.65	0.507
Disease duration	9.11 ± 6.83	9.83 ± 3.29	0.770

^a^ Fisher’s exact test.

**Table 4 ijms-24-06377-t004:** Detailed values of the comparison of AD and CuAD neuropsychological test battery.

	AD	CuAD	*p* Value	Adjusted *p* Value ^b^	B-H Adjusted *p* Value ^c^
MMSE	15.11 ± 8.33	8.39 ± 6.10	0.025	0.069	0.115
MOCA	10.11 ± 7.66	5.39 ± 6.51	0.106	0.183	0.229
FAB	8.00 ± 5.10	2.61 ± 3.07	0.012 ^a^	0.008	0.040
CDT	4.33 ± 2.96	1.89 ± 1.84	0.037 ^a^	0.029	0.072
NPI	17.44 ± 12.41	19.39 ± 14.52	0.734	0.556	0.556

^a^ Welch’s *t*-test. ^b^ an ANCOVA model was run to assess the effect of diagnosis adjusted for sex and age. ^c^ B-H: Benjamini & Hochberg’s *p* value adjustment for multiple testing [34].

**Table 5 ijms-24-06377-t005:** Correlations between neuropsychological test battery outcomes and Cu biomarkers.

		Cu	Cp	Cp Activity	Cu:Cp	Non-Cp Cu	eCp:iCp
MMSE	Pearson Correlation	−0.343	−0.205	−0.370	−0.379	−0.412	−0.384
	*p* value	0.079	0.305	0.058	0.051	0.029	0.048
	B-H adjusted p ^a^	0.095	0.305	0.087	0.087	0.087	0.087
MOCA	Pearson Correlation	−0.254	−0.059	−0.248	−0.383	−0.376	−0.291
	*p* value	0.256	0.770	0.256	0.147	0.147	0.256
	B-H adjusted p ^a^	0.307	0.770	0.307	0.307	0.307	0.307
FAB	Pearson Correlation	−0.437	−0.296	−0.424	−0.478	−0.476	−0.417
	*p* value	0.023	0.134	0.028	0.012	0.010	0.030
	B-H adjusted p ^a^	0.036	0.134	0.036	0.036	0.036	0.036
CDT	Pearson Correlation	−0.328	−0.245	−0.363	−0.330	−0.352	−0.360
	*p* value	0.095	0.218	0.063	0.093	0.066	0.066
	B-H adjusted p ^a^	0.114	0.218	0.114	0.114	0.114	0.114
NPI	Pearson Correlation	0.003	−0.104	−0.049	0.143	0.039	0.000
	*p* value	0.990	0.605	0.810	0.478	0.843	1.000
	B-H adjusted p ^a^	0.999	0.999	0.999	0.999	0.990	0.999

^a^ B-H: Benjamini & Hochberg’s *p* value adjustment for multiple testing [34]; eCp:iCp: ceruloplasmin specific activity.

**Table 6 ijms-24-06377-t006:** Biochemical features of the CuAD patient.

Test	Result	Normal Reference Values
Ferritin	230.3	24–307 ng/mL
γ-GT	11	3–29 U/L
AST	15	2–40 U/L
ALT	8	2–40 U/L
Magnesium	1.84	1.3–2.1 mg/dL
Phosphorus	2.6	2.8 to 4.5 mg/dL
Lithium	<0.0101	0.6 and 1.2 mmol/L
Bilirubin total	0.3	0–1.1 mg/dL
Iron (Fe)	31	37–164 µg/dL
Transferrin	187	200–360 mg/dL
TIBC ^&^	233.8	240 to 450 mg/dL
Transferrin saturation (%)	13.3	15–50%
Copper (Cu)	5	11.0–24.0 µM
Ceruloplasmin	24.23	20–60 mg/dl
Cp activity	232	62–140 IU
Non-Cp Cu *	4	0.05–1.6 µg/dL

^&^ TIBC—total iron binding capacity; * For non-Cp Cu measurements, serum samples were filtered through an Amicon Ultra centrifugal filter unit according to previously published methods [36]; γ-GT—Gamma-glutamyl transferase; AST—Aspartate transaminase; ALT—Alanine transaminase.

## Data Availability

Data supporting reported results can be provided by contacting the corresponding author.

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
