# Peer review of "Non-Ceruloplasmin Copper Identifies a Subtype of Alzheimer’s Disease (CuAD): Characterization of the Cognitive Profile and Case of a CuAD Patient Carrying an RGS7 Stop-Loss Variant"

_ijms, 2023, doi:10.3390/ijms24076377_

Round 1
Reviewer 1 Report (New Reviewer)
Squitti et al. provided original manuscript entitled “Non-ceruloplasmin copper identifies a subtype of Alzheimer’s disease (CuAD): characterization of the cognitive profile and case of a CuAD patient carrying a RGS7 stop-loss variant”. It is well designed and well written article with interesting FDG PET from cuAD patient, however there are few concerns that need to be addressed.
Major concerns,
Does all the patients (both AD and CuAD) diagnosed for the CSF markers? Or PET markers?
1) I would like to see the correlation between the CSF/PET markers (Aβ42, T-tau, P-tau) Vs each of the Cu panel (Cu, non-Cp Cu, ceruloplasmin, ceruloplasmin and ceruloplasmin specific activity)
2) Also, I would like to see the correlation between the neuropsychological test battery Vs each of the Cu panel (Cu, non-Cp Cu, ceruloplasmin, ceruloplasmin and ceruloplasmin specific activity)
3) In figure 1, colour the data points within AD category to show distinction for AD and CuAD patients.
4) Claims about RGS7 in some places overstated such as “supporting the role of the RGS7 protein in regulating Cu trafficking” without any experimental evidence.
5) The study features excessive self-citation. Studies by other groups should be cited, appropriately.
Minor,
Figure 7, 8, 9 can be combined into single figure.
Use full name for abbreviation. Such as IGV in figure 4. Check for all abbreviations.
In line 114 check for typo – “Twenty-8”
In line 125 check for phrase – “ AD one”
Author Response
Reviewer 1
Squitti et al. provided original manuscript entitled “Non-ceruloplasmin copper identifies a subtype of Alzheimer’s disease (CuAD): characterization of the cognitive profile and case of a CuAD patient carrying a RGS7 stop-loss variant”. It is well designed and well written article with interesting FDG PET from cuAD patient, however there are few concerns that need to be addressed.
Major concerns,
Does all the patients (both AD and CuAD) diagnosed for the CSF markers? Or PET markers?
Reply: yes, patients received an IGW2 criteria Alzheimer’s disease diagnosis, which implies a marked reduction in cerebrospinal (CSF) Aβ42 and in the Aβ42/Aβ40 ratio, increased levels of P-tau or T-tau in the CSF, or a positive amyloid positron emission tomography (PET) or fluorodeoxyglucose (FDG)- PET; see Methods section p. 15, ll, 434-430
1) I would like to see the correlation between the CSF/PET markers (Aβ42, T-tau, P-tau) Vs each of the Cu panel (Cu, non-Cp Cu, ceruloplasmin, ceruloplasmin and ceruloplasmin specific activity)
Reply: As per the reply to the first comment, patients received an IGW2 diagnosis. However, to fulfil the criterion of Alzheimer’s disease biomarkers positivity in CSF/brain, patients underwent diverse analyses: CSF AD biomarkers assessment or alternatively Aβ-PET or FDG-PET scan, and it was not ethically correct to ask them for a further invasive (lumbar puncture) or expensive (PET scan) analysis, only for the sake of the study
2) Also, I would like to see the correlation between the neuropsychological test battery Vs each of the Cu panel (Cu, non-Cp Cu, ceruloplasmin, ceruloplasmin and ceruloplasmin specific activity)
Reply: correlations between the neuropsychological test battery Vs each of the Cu panel biomarkers are provided in the new version of the manuscript. We applied Benjamini & Hochberg’s correction to control for the false discovery rate. Worse scores on the FAB scale scores resulted associated with abnormal values of the Cu biomarkers. A new Table 5 has been added in the new version of the manuscript, as well as new text Methods (p. 15, ll. 481-484); Results (p. 6, ll. 193-198); Discussion (p. 13, ll. 332-334)
3) In figure 1, colour the data points within the AD category to show the distinction between AD and CuAD patients.
Reply: a new Figure 2, showing by different colours AD (black circles) and CuAD (grey circles) has been provided
4) Claims about RGS7 in some places overstated such as “supporting the role of the RGS7 protein in regulating Cu trafficking” without any experimental evidence.
Reply: we toned down our statements. The paragraph now reads: ‘Another result is the identification of an RGS7 stop-loss variant in a CuAD patient with severely abnormal Cu, supporting a potential role of the RGS7 protein in regulating Cu trafficking’ (p. 12, ll. 320-321) and again “…However, the data presented are merely descriptive and further experimental evidence is needed to claim a causative connection between the RGS7 stop-loss variant and the AD onset or progression, as well as the role of RGS7 as a protein involved in regulating Cu trafficking…” (p. 12, ll. 324-325)
5) The study features excessive self-citation. Studies by other groups should be cited, appropriately.
Reply: we cut the number of self-citations as requested. In the new version, only 3 references of 62 quoted were from our group. New references have been quoted
Minor,
Figure 7, 8, 9 can be combined into a single figure.
Reply: As suggested we combined figure 7,8 and 9 in a sole new Figure 7. The Legend and the text was modified consequently
Use full name for abbreviation. Such as IGV in figure 4. Check for all abbreviations.
Reply: IGV has been substituted with “Integrative genomics viewer”, thanks for noting that
In line 114 check for typo – “Twenty-8”
Reply: we replaced “Twenty-8” with “Twenty-eight”
In line 125 check for phrase – “ AD one”
Reply: we replaced “ AD one” with “AD group”
Reviewer 2 Report (New Reviewer)
Cu is an essential metal necessary for brain development and physiology. Severe Cu deficiencies are associated with immune, cardiac, bone, and central nervous system conditions. Cu absorption, distribution, and homeostasis in the brain are tightly controlled, with the neurovascular unit and thee blood-brain barrier playing an essential role in the process. Mutations in the genes encoding for proteins involved in the Cu pathway results in several hereditary diseases. Many findings indicate that Cu dyshomeostasis plays a critical part in AD. The metal has a direct role in amyloid pathology by promoting Aβ aggregation.
The aim of this study was a new pilot diagnostic pathway to improve the diagnostic accuracy of AD, based on non-Cp-Cu as a stratification add-on biomarker. About 75–95% of total serum Cu binds strongly and inertly to ceruloplasmin, while about 5–10% circulates in a weaker and more labile form, being exchanged among various protein compounds. On this basis, these Cu complexes have been defined as non-ceruloplasmin Cu (non-Cp Cu), a clinical biomarker applied to Wilson’s disease (WD), a paradigmatic disorder of Cu toxicosis and accumulation.
Furthermore, patients with abnormal values in at least two biomarkers of the Cu panel underwent whole exome sequencing (WES) analysis to screen for potential causative gene variants.
In my opinion, the manuscript is clear, relevant to the field, and presented in a well-structured manner. The manuscript is scientifically substantiated, the hypotheses that have been formulated find their application in the literature, and the experimental design is appropriate to test the hypothesis. The applied methodology and its description in the manuscript can be used to confirm the obtained results. Figures, tables and diagrams are presented clearly and correctly interpreted. The statistical methods used: the Kolmogorov-Smirnov test (p>0.05), chi-square, Fisher's exact test, Benjamini and Hochberg's correction confirm the value of the obtained results. The presented conclusions regarding the role of Cu in AD, the mechanism of action and the hypothesis of genetic background as well as the potential role of the research panel in the AD prevention strategy are justified by the presented research results. The study was approved by Local Ethics Committee 568 (approval no. 1/1999).
In my opinion, the article can be adopted without further changes.
Author Response
Reply: we thank the reviewer for the positive consideration of our work
Round 2
Reviewer 1 Report (New Reviewer)
Authors have responded to the comments. I don't have major concerns
Still figure quality and panel labelling can improved ( for example Figure 7)
This manuscript is a resubmission of an earlier submission. The following is a list of the peer review reports and author responses from that submission.
Round 1
Reviewer 1 Report
The paper is significant as it explores alternative diagnostic approaches of AD, but the writing is not clear, many unclear methodological issues, statistical tests and results do not allow the conclusions to be drawn. Below I have detailed my questions:
Introduction
1. What is the relationship between Cu levels and Aβ and tau levels, in the brain and in the serum, if there is any relevant data in the literature?
2. More background about RGS7 stop-loss variant
Methods: I found the methods in the paper to be unclear and insufficient to help me understand the details of the design and stats analysis.
Measures:
3. Put some background regarding why using 1.6 mol/L as a cut-off
4. In line 353, why is there a decreased level of Aβ-42 in AD patients? Shouldn’t there increased level of it?
5. In line 127, why is Cu:Cp ratio a relevant measure? Please explain.
6. In figure 1, there needed more background about the different ceruloplasmin measures.
Statistical analyses:
7. In line 125, why is a Welch’s t-test used rather than a regular 2-sample t-test? In line 149, why Mann Whitney test? justify why using a non-parametric test? is it the data distribution or why? There is brief description of the stats analysis process in 4.3. Statistical Analysis, but the actual distribution results should be provided so the readers could also make sense of the decisions about which stat analyses was picked.
8. In figure 1, are the scatterplots showing data that is residualized for age and sex? If so, state that the tests are controlled for individual differences in age and sex so it is clear. If not, clarify that the data plotted are not controlled for age and sex. Same for Figure 2.
9. Table 4, line 152, “ a Fisher's exact test.” but I don’t see an annotation of “a” or where Fisher’s exact test would have been conducted for the comparisons in table 4.
10. When comparing multiple cognitive abilities between AD and CuAD, what kind of correction was conducted to avoid false positive due to multiple testing?
Study Design:
11. I am not sure why is the case study presented. Why is there only one data point showing significantly Cu level and also a RGS7 stop-loss variant? Anyone else?
Conclusion:
12. The authors used the results that CuAD patients scored significantly lower on FAB and CDT tests to support the executive function deficits in CuAD. This interpretation may be false:
a. when correcting for multiple testing
b. MMSE and MOCA also show similar pattern but not significant, the group difference in these tests may not be significantly different from the group difference in FAB and CDT. ANOVA testing the interaction of patient group (between-subject: CuAD vs AD) and neuropsychological tests (within-subject variable) should be conducted, and only a significant interaction could support that a cognitive deficit in CuAD specific to executive function.
Other:
13. Language and writing needs improvement, below are just some examples:
14. Line 387: “our hypothesis was to explore” – hypothesis should be a statement rather than to explore something.
15. Line 388, there seems to have a comma missed.
16. In line 136, “b an ANCOVA model was run to assess the effect of diagnosis adjusted for sex and 136 age.” I didn’t see “b” in the table.
Author Response
Reviewer 1
The paper is significant as it explores alternative diagnostic approaches of AD, but the writing is not clear, many unclear methodological issues, statistical tests, and results do not allow conclusions to be drawn. Below I have detailed my questions:
Reply: we thank the reviewer for the comment. We have added information about methods and statistical tests and addressed the issues about results. On the basis of these clarifications, we hope that the manuscript is now suitable for publication. We have provided a line-by-line reply with reference to the changes in the revised text that has been also yellow-highlighted for better clarity. The text has also English edited.
Introduction
- What is the relationship between Cu levels and Aβ and tau levels, in the brain and in the serum, if there is any relevant data in the literature?
Reply: there is massive data in this regard that are summarized in two recent reviews entailed Copper Imbalance in Alzheimer's Disease and Its Link with the Amyloid Hypothesis: Towards a Combined Clinical, Chemical, and Genetic Etiology.Squitti R, Faller P, Hureau C, Granzotto A, White AR, Kepp KP.J Alzheimers Dis. 2021;83(1):23-41. doi: 10.3233/JAD-201556. And Kepp KP and Squitti R APP “Copper imbalance in Alzheimer’s disease: Convergence of the chemistry and the clinic” Coordinatin Chemistry Review 2019 that we quoted in the manuscript
Summarizing: Cu has a direct role in amyloid pathology by promoting Aβ aggregation; Aβ can trigger neurotoxic effects by promoting deficits of intracellular Cu ([Cu]i); very importantly, it has been found that AβPP, from which Aβ is produced, can bind Cu(I) and Cu(II) with picomolar affinity in vitro [1, 2]. AβPP-KO mice exhibit increased Cu levels in the cerebral cortex, whereas the over-expression of AβPP leads to significantly reduced brain levels of Cu in a preclinical model of AD [3]. Moreover, Cu levels can also significantly affect the neuronal redox state, thereby indicating a pathogenic link between Aβ dysmetabolism, oxidative stress, and Cu dyshomeostasis.
Regarding the link of Aβ and Cu in serum, a previous paper of ours indicated a correlation between the two (Squitti R, et al Excess of nonceruloplasmin serum copper in AD correlates with MMSE, CSF [beta]-amyloid, and h-tau..Neurology. 2006 Jul 11;67(1):76-82. doi: 10.1212/01.wnl.0000223343.82809.cf.PMID: 16832081).
Some of these concepts have now been added in the new version of the manuscript (Lines 70-91)
- More background about RGS7 stop-loss variant
Reply: Actually, there is no published literature that could provide more background about this specific stop-loss variant. In the article we synthesized the key concepts about RGS7 protein according to published papers, highlighting that it's a protein especially expressed in the frontal cortex, we mentioned its principal known function, its main interactors, and their involvement in diseases with neurodegeneration as the main mechanism of disease like Parkinson and Lewy body Dementia and we provided in-silico analysis to demonstrate at our best a biologically significant alteration of mutated vs wild-type protein. We also described the frequency of the stop-loss variant in RGS7 across the general population and speculated about possible dominant vs. recessive eventual action based on its frequency. To the best of our knowledge, it's not possible to add more background, unless we start to speculate about molecular mechanisms causing possible disease (i.e. accumulation of altered protein in neural cells or dominant-negative effect caused by loss-stop variant), but that would go very beyond the scope of the article. This information is reported in lines 353-364.
Methods: I found the methods in the paper to be unclear and insufficient to help me understand the details of the design and stats analysis.
Reply: The reviewer is correct in asking for clarification since we have condensed in a pilot diagnostic pathway, diverse sub-studies, all devoted to the demonstration of the diagnostic utility of the add-on test non-Cp Cu. Specifically, we adopted an add-on test to further enhance the diagnostic accuracy of AD and identify additional disease-associated metabolic sub-pathways, in this case, related to Cu abnormalities. The add-on test has been applied after the standard diagnostic pathway based on IGW2 criteria has been completed, and has been employed as a stratification biomarker to identify subgroups of patients.
We have previously published 3 studies devoted to the demonstration of the existence of a copper subtype of AD that shared a similar methodological design [4-6]. In general, terms, to demonstrate the existence of subtypes of disease four strategies can be applied: 1.identification of a bimodal distribution of the biomarker of interest in the patient population; 2.the response to a specific therapy; 3.cluster analysis; 4.the identification of a specific biomarker that distinguishes the cases, classifying them in sub-groups that are then compared for demographic, clinical, genetic, or biological variables. If groups are different and heterogeneity between them can be detected, then this speaks in favor of disease subtypes [7]. As in the above-mentioned studies of ours [4-6] we applied the latter strategy, employing, in this case, scores from neuropsychological testing for the comparison between the 2 groups.
Preliminarily, to show that this new AD population, on average, has abnormal values of Cu, we compared them to healthy controls (section 2.1). At this point, we applied the 1.6 µmol/L cut-off at the sole AD population, and this clearly identified two subgroups: AD (non-Cp Cu < 1.6 µmol/L) and CuAD (non-Cp Cu ≥ 1.6 µmol/L) (Section 2.2).
Furthermore, to complete the pilot diagnostic path, in case of abnormalities (out of normal reference range) of at least two biomarkers of the Cu biomarkers panel, whole exome sequencing (WES) analysis was applied to screen for potential genetic causative variants that can explain the strong Cu imbalance observed. Identifying potential genetic variants causative of the Cu imbalance may support the application of anti-Cu therapeutic options out of the label, as per the precision medicine principles: delivery of individually adapted medical care based on the genetic characteristics of each patient [8] (Section 2.3).
The pilot study proposed, then, is a small-scale application of a diagnostic study devoted to the identification of additional disease-associated metabolic sub-pathways employing an add-on test, to support the identification of causative genetic variants - by means of a WES approach - that can explain the metabolic imbalance observed and that might help to identify therapeutic approach out of the label, based on the genetic variant, to curb the additional disease-associated metabolic imbalance as per the precision medicine principle. However, in the current study, we didn’t identify mutations in established genes that specify for Cu proteins (i.e. CTR1, ATP7B, ATP7A, COMMD1, ATOX1) as we expected, but a stop-loss variant in the RGS7 gene. The causative association among RGS7, Cu abnormal metabolism, and AD development cannot be claimed at this stage of our research, since additional evidence is needed. Nevertheless, we feel that data can be presented to the readers to rise interest in the topic and stimulate research. We are actually in the process of a cross-sectional study exploring the frequency of the RGS7 mutation in an AD population vs. healthy controls.
In brief, we developed a pilot diagnostic study in AD that can be summarized as follows:
- Standard diagnostic pathway application (IGW2 criteria);
- Add-on test application to stratify the AD population;
- Search for genetic variants that might explain the disease-associated metabolic sub-pathways (we limited the WES analysis to patients with at least two biomarkers of the Cu panel altered).
The pilot diagnostic path developed is eventually devoted to the aim of administrating (future application):
- Therapeutic treatments are out of label to curb the metabolic imbalance based on precision medicine principles.
To accomplish reviewer clarification about the methods of the current investigation, we added a paragraph in the Methods that included some of these concepts in the new version of the manuscript (pp. 14-15; lines 444-464).
Regarding the statistical methods, we added a new text (pages 15, lines 501-511)
Measures:
- Put some background regarding why using 1.6 mol/L as a cut-off
Reply: Non-Cp Cu is an established biomarker for Wilson disease [9-11]. A cut-off of 1.6 μM has been established on the basis of previous experience in WD clinical practice [9-11]. We recently discussed this topic in [12] presenting in Table 3 of that paper [12] a summary of the non-Cp Cu cut-offs. The bulk of evidence collected allows to use of non-Cp Cu cut-off values in discriminating AD patients from healthy subjects, Wilson disease patients from healthy individuals, or stable MCI subjects from MCI converting to AD (Table 3). People with values of non-Cp Cu in serum lower than 1.6 µmol/L after an overnight fast are in the normal range [9-12]. Increased values can be found in AD patients pertaining to the CuAD subset, while values higher than 2.31 µmol/L most likely pertain to WD patients.
These concepts have been added in the new version of the manuscript (lines 86-89)
- In line 353, why is there a decreased level of Aβ-42 in AD patients? Shouldn’t there increased level of it?
Reply: Alzheimer's disease is thought to be caused by the abnormal build-up of proteins in and around brain cells. One of the proteins involved is called amyloid, deposits of which form plaques around brain cells. The other protein is called tau, deposits of which form tangles within brain cells. One form, amyloid-β 42, is thought to be especially toxic. In the Alzheimer's brain, abnormal levels of this naturally occurring protein clump together to form plaques that collect between neurons and disrupt cell function.
Accumulation of aggregated amyloid-β 42 in the brain is believed to be the first pathological mechanism of Alzheimer's disease. In cerebrospinal fluid, the changes observed include a 50% reduction of amyloid-β 42 as a consequence of amyloid deposition in the brain.
- In line 127, why is Cu:Cp ratio a relevant measure? Please explain.
Reply: Copper: Ceruloplasmin (Cu:Cp) is an additional copper index that serves as a useful internal quality control verification of ceruloplasmin calibration, and provides information about the actual stoichiometry between copper and ceruloplasmin in the specimens. From the knowledge that the correct stoichiometry for copper and ceruloplasmin is 6-8 atoms of copper per ceruloplasmin molecule [13-15], the plausible theoretical values of Cu:Cp which should be effectively measured in specimens of healthy controls should range between 6-8, even though this ratio can practically yield diverse values.
We use the Cu:Cp ratio as an index to verify how the actual stoichiometry measured in the healthy control specimens is close to the theoretically correct one.
A Cu:Cp value of 6.6 has been advocated as the theoretical optimal ratio for healthy subjects [16]. This value has been confirmed in a previous meta-analysis [17]. The Cu:Cp study in healthy controls of a more recent study [18] revealed a value of 6.5 (1.0). The Cu:Cp ratio in healthy controls of the current study is 6.4 ± 0.65 (Table 2), indicating that copper and ceruloplasmin in the specimens analyzed are close to the correct stoichiometry value [16-18]. We added some clarification in the Methods section, (page 15, lines 490-493), since all the information is available in previous literature [16-18].
- In figure 1, there needed more background about the different ceruloplasmin measures.
Reply: We added some information in the Figure legend of Figure 2 (lines 159-162)
Statistical analyses:
- In line 125, why is a Welch’s t-test used rather than a regular 2-sample t-test? In line 149, why Mann Whitney test? justify why using a non-parametric test? is it the data distribution or why? There is brief description of the stats analysis process in 4.3. Statistical Analysis, but the actual distribution results should be provided so the readers could also make sense of the decisions about which stat analyses was picked.
Reply: We thank the reviewer who noted that the Mann-Whitney test in line 149 was improperly quoted, it was a typo, and we delete it. We re-wrote the statistical methods specifying when and why we used a Welch’s test rather than a regular T-test (page 15- lines 503-507).
- In figure 1, are the scatterplots showing data that is residualized for age and sex? If so, state that the tests are controlled for individual differences in age and sex so it is clear. If not, clarify that the data plotted are not controlled for age and sex. Same for Figure 2.
Reply: data in the Figure 1 are raw data not adjusted for subjects’ differences in sex and age. This has been now stated in the new version of the Figure legends (lines 162-163).
- Table 4, line 152, “ a Fisher's exact test.” but I don’t see an annotation of “a” or where Fisher’s exact test would have been conducted for the comparisons in table 4.
Reply: It was a typo, we corrected the annotation in Table 4 and specified the right test, Welch’s t-test
- When comparing multiple cognitive abilities between AD and CuAD, what kind of correction was conducted to avoid false positive due to multiple testing?
Reply: As requested we added a Benjamini & Hochberg p value adjustment for multiple testing, a new Table 4 was proposed
Study Design:
- I am not sure why is the case study presented. Why is there only one data point showing a significantly Cu level and also an RGS7 stop-loss variant? Anyone else?
Reply: We clarified in a previous reply to rev1 the study design. This is a small-scale application of a diagnostic study devoted to the identification of additional disease-associated metabolic sub-pathways employing an add-on test, and to the identification of the potential causative genetic variant - by means of a WES approach - that might explain the metabolic imbalance observed this has been summarized in the new version of the methods (lines 444-464). As reported in the original submission (at line 133), only in case of at least two biomarkers of the Cu panel altered, patients underwent WES (line 129). At this stage of our research, we have identified only a patient with these features.
Conclusion:
- The authors used the results that CuAD patients scored significantly lower on FAB and CDT tests to support the executive function deficits in CuAD. This interpretation may be false:
- when correcting for multiple testing
We thank the reviewer for this suggestion. On the basis of the Benjamini & Hochberg p-value adjustment for multiple testing in Table 4, only the FAB resulted significantly different between the two AD groups. We have then corrected our results and conclusion (page 5, lines 182-187, Table 4, page 12, lines 331-337, page 15, lines 504-509, and page 17, line 582 the term CDT has been deleted).
- MMSE and MOCA also show similar pattern but not significant, the group difference in these tests may not be significantly different from the group difference in FAB and CDT. ANOVA testing the interaction of patient group (between-subject: CuAD vs AD) and neuropsychological tests (within-subject variable) should be conducted, and only a significant interaction could support that a cognitive deficit in CuAD specific to executive function.
Reply: We referred to a previous reference study (PMID: 15262742) that evaluated if the FBA could contribute to the differential diagnosis of two types of dementia, similar to what we tried to achieve in our pilot study. In that reference study (PMID: 15262742), the authors did not use the repeated measure ANOVA. Furthermore, the cognitive scales we used in the current study have different ranges (FAB has a range of 0-18; MOCA has a range of 0-30; MMSE has a range of 0-30; NPI has a range of 0-14; CDT has a range 0-61) that prevent the application of the repeated measure ANOVA. Thus, we thank the reviewer for the suggestion, but we feel that the repeated measure ANOVA is more suitable for a clinical trial study in which the same index is measured under different conditions so the resulting index's measures are confrontable [e.g. in clinical trial interventions: Sham vs. electrical stimulation or Investigational Medicinal Product (IMP in other words, a new drug) vs. placebo or vs. a Gold standard medication]. However, to accomplish the reviewer’s request for more evidence of the significance of the difference between the two groups, we calculated the ‘d’ Cohen (added also in the Methods, Page 16, lines 507-510). Cohen's d’ can provide a standardized measure of the size of the difference between groups. Cohen’s d demonstrates a large difference between the ‘normal AD’ and the ‘CuAD’ for MMSE (d=0.84, 95%CI: 0.01; 1.68), FBA (d=1.24, 95%CI: 0.37; 2.11), and CDT (d=0.99, 95%CI: 0.15; 1.85), while it is medium for MOCA (d=0.59, 95%CI: -0.23; 1.41) and small for NPI (d=0.26, 95%CI: -1.06; 0.55) (page 5, lines 185-188)
Other:
- Language and writing needs improvement, below are just some examples:
Reply: we have accomplished an English editing
- Line 387: “our hypothesis was to explore” – hypothesis should be a statement rather than to explore something.
Reply: we thank the reviewer: we improved the sentence.
- Line 388, there seems to have a comma missed.
Reply: fixed
- In line 136, “b an ANCOVA model was run to assess the effect of diagnosis adjusted for sex and 136 age.” I didn’t see “b” in the table.
- Reply: the ‘b’ has been added (page 4, line 164)
Reference for reviewer’s clarification
- Young, T. R.; Pukala, T. L.; Cappai, R.; Wedd, A. G.; Xiao, Z., The Human Amyloid Precursor Protein Binds Copper Ions Dominated by a Picomolar-Affinity Site in the Helix-Rich E2 Domain. Biochemistry 2018, 57, (28), 4165-4176.
- Young, T. R.; Wedd, A. G.; Xiao, Z., Evaluation of Cu(i) binding to the E2 domain of the amyloid precursor protein - a lesson in quantification of metal binding to proteins via ligand competition. Metallomics 2018, 10, (1), 108-119.
- White, A. R.; Bush, A. I.; Beyreuther, K.; Masters, C. L.; Cappai, R., Exacerbation of copper toxicity in primary neuronal cultures depleted of cellular glutathione. J Neurochem 1999, 72, (5), 2092-8.
- Squitti, R.; Simonelli, I.; Cassetta, E.; Lupoi, D.; Rongioletti, M.; Ventriglia, M.; Siotto, M., Patients with Increased Non-Ceruloplasmin Copper Appear a Distinct Sub-Group of Alzheimer's Disease: a Neuroimaging Study. Curr Alzheimer Res 2017.
- Squitti, R.; Ventriglia, M.; Gennarelli, M.; Colabufo, N. A.; El Idrissi, I. G.; Bucossi, S.; Mariani, S.; Rongioletti, M.; Zanetti, O.; Congiu, C.; Rossini, P. M.; Bonvicini, C., Non-Ceruloplasmin Copper Distincts Subtypes in Alzheimer's Disease: a Genetic Study of ATP7B Frequency. Molecular neurobiology 2017, 54, (1), 671-681.
- Tecchio, F.; Vecchio, F.; Ventriglia, M.; Porcaro, C.; Miraglia, F.; Siotto, M.; Rossini, P. M.; Rongioletti, M.; Squitti, R., Non-Ceruloplasmin Copper Distinguishes A Distinct Subtype of Alzheimer`s Disease: A Study of EEG-Derived Brain Activity. Curr Alzheimer Res 2016.
- Murray, M. E.; Graff-Radford, N. R.; Ross, O. A.; Petersen, R. C.; Duara, R.; Dickson, D. W., Neuropathologically defined subtypes of Alzheimer's disease with distinct clinical characteristics: a retrospective study. Lancet Neurol 2011, 10, (9), 785-96.
- Servant, N.; Romejon, J.; Gestraud, P.; La Rosa, P.; Lucotte, G.; Lair, S.; Bernard, V.; Zeitouni, B.; Coffin, F.; Jules-Clement, G.; Yvon, F.; Lermine, A.; Poullet, P.; Liva, S.; Pook, S.; Popova, T.; Barette, C.; Prud'homme, F.; Dick, J. G.; Kamal, M.; Le Tourneau, C.; Barillot, E.; Hupe, P., Bioinformatics for precision medicine in oncology: principles and application to the SHIVA clinical trial. Front Genet 2014, 5, 152.
- Roberts, E. A.; Schilsky, M. L.; American Association for Study of Liver, D., Diagnosis and treatment of Wilson disease: an update. Hepatology 2008, 47, (6), 2089-111.
- Walshe, J. M.; Clinical Investigations Standing Committee of the Association of Clinical, B., Wilson's disease: the importance of measuring serum caeruloplasmin non-immunologically. Ann Clin Biochem 2003, 40, (Pt 2), 115-21.
- Hoogenraad, T., Wilson disease. 2001.
- Squitti, R.; Ventriglia, M.; Granzotto, A.; Sensi, S. L.; Rongioletti, M. C. A., Non-Ceruloplasmin Copper as a Stratification Biomarker of Alzheimer's Disease Patients: How to Measure and Use It. Curr Alzheimer Res 2021, 18, (7), 533-545.
- Bento, I.; Peixoto, C.; Zaitsev, V. N.; Lindley, P. F., Ceruloplasmin revisited: structural and functional roles of various metal cation-binding sites. Acta Crystallogr D Biol Crystallogr 2007, 63, (Pt 2), 240-8.
- Bielli, P.; Calabrese, L., Structure to function relationships in ceruloplasmin: a 'moonlighting' protein. Cell Mol Life Sci 2002, 59, (9), 1413-27.
- Lopez-Avila, V.; Sharpe, O.; Robinson, W. H., Determination of ceruloplasmin in human serum by SEC-ICPMS. Anal Bioanal Chem 2006, 386, (1), 180-7.
- Twomey, P. J.; Viljoen, A.; House, I. M.; Reynolds, T. M.; Wierzbicki, A. S., Copper:caeruloplasmin ratio. J Clin Pathol 2007, 60, (4), 441-2.
- Squitti, R.; Simonelli, I.; Ventriglia, M.; Siotto, M.; Pasqualetti, P.; Rembach, A.; Doecke, J.; Bush, A. I., Meta-analysis of serum non-ceruloplasmin copper in Alzheimer's disease. J Alzheimers Dis 2014, 38, (4), 809-22.
- Squitti, R.; Ghidoni, R.; Simonelli, I.; Ivanova, I. D.; Colabufo, N. A.; Zuin, M.; Benussi, L.; Binetti, G.; Cassetta, E.; Rongioletti, M.; Siotto, M., Copper dyshomeostasis in Wilson disease and Alzheimer's disease as shown by serum and urine copper indicators. J Trace Elem Med Biol 2018, 45, 181-188.
- Squitti, R.; Siotto, M.; Cassetta, E.; Idrissi, I. G.; Colabufo, N. A., Measurements of serum non-ceruloplasmin copper by a direct fluorescent method specific to Cu(II). Clin Chem Lab Med 2017, 55, (9), 1360 -1367.
Reviewer 2 Report
Here the authors report an interesting finding of a AD subgroup that is denoted by a marked increase in levels of Non-Cp Cu levels in blood plasma. While the work certainly deserves to be published, I would like some of my concerns addressed prior to that:
1) My main concern with this manuscript is that while its striking that the Non-Cp Cu levels serve as a strong biomarker for a AD-subgroup which also correlates with decreased performance in executive function outcomes, the linking of the RGS7 stop-loss mutation genotype to the Non-Cp Cu high phenotype is very weak. This is due to the fact that the exome analysis was only conducted in one patient. I would strongly suggest the authors to tone done the language of this link in the manuscript until further experiments support the notion that this RGS7 mutation is found in the Non-Cp Cu high AD subgroup patients exclusively. If such data exists at this point, including those results will really bolster such linking.
2) In figure 1, why was z-scores used instead of absolute values ?
3) The results discussed in line 143-144 must be included (at least in supplementary results) as its central to the main findings of the manuscript.
4) Figure 2 should be represented as a beeswarm plot like Figure 1.
5) Figure 5 is very unclear. It should include details such as (not limiting to): a) species of the sequences compared clearly labeled with identifiers possibly in parenthesis, b) the 19 and/or 14 residues of the extended protein that is conserved should be highlighted clearly (maybe using a box), c) what is the extent of the conservation ? Is it conserved only in mammals or vertebrates or beyond ?
6) Line 246 says "Another result identifies the role of the RGS7 protein in regulating Cu trafficking". This statement is not substantiated as there is no functional link established in relation to RGS7 in the results of this manuscript. If such a role is known, please elaborate. This is akin to my concern mentioned in my first point.
Other minor corrections:
1) line 280 "lowy body" is a typo.
2) line 349 "were recruited" repeated twice.
3) line 350 "Twenty-8" is a typo.
4) A full stop is missing in line 388.
5) Figure 3 is very blurry. Please consider uploading a higher resolution figure.
Author Response
Reviewer 2
Comments and Suggestions for Authors
Here the authors report an interesting finding of an AD subgroup that is denoted by a marked increase in levels of Non-Cp Cu levels in blood plasma. While the work certainly deserves to be published, I would like some of my concerns addressed prior to that:
1) My main concern with this manuscript is that while its striking that the Non-Cp Cu levels serve as a strong biomarker for a AD-subgroup which also correlates with decreased performance in executive function outcomes, the linking of the RGS7 stop-loss mutation genotype to the Non-Cp Cu high phenotype is very weak. This is due to the fact that the exome analysis was only conducted in one patient. I would strongly suggest the authors to tone done the language of this link in the manuscript until further experiments support the notion that this RGS7 mutation is found in the Non-Cp Cu high AD subgroup patients exclusively. If such data exists at this point, including those results will really bolster such linking.
Reply: Unfortunately, we have no access to new biological material from the patient, so all the useful studies you correctly suggest (i.e. in vivo colocalization of altered protein, expression studies, and in vivo alteration of functional pathways) are not available at the moment, but we are performing other functional studies and trying to recreate in vivo models with the same mutation. Furthermore, we are in progress with a study evaluating the frequency of the RGS7 mutations in a CuAD population. The reviewer is definitely right about the frail link between non-ceruloplasmin Cu phenotype and RGS7 mutation with these data, that's why we carefully avoided any kind of expression stating that RGS7 is the cause of that phenotype. Of course, we "intuitively" think that there is a link between the RGS7 mutation and the abnormal Cu biomarkers of the patient with AD since no other mutations have been identified that can explain the peculiar copper markers abnormalities that prompted us to search for mutations (see also reply to rev 1 comments). However, we surely agree that more experimental data are needed to clearly demonstrate a causative connection. Due to actual experimental limitations, the scope of this article is merely descriptive (see lines 327-329 of the new version), due to the striking observation of the co-occurrence of a particular biochemical phenotype that can characterize a subgroup of Alzheimer's disease and the presence of homozygous mutation which it's hard to think to not have a biological impact. As discussed in other comments (to reviewer 1) and better detailed in the new version of the methods, we intend to provide a small-scale application of a diagnostic path devoted to the identification of additional disease-associated metabolic sub-pathways employing an add-on test, and to the identification of the potential causative genetic variant - by means of a WES approach – but we are well aware that we are still far from establishing a causative connection between RGS7 mutation and AD.
So, although we provide a basis to justify the connection between RGS7 and at least AD, we consider that the language is appropriate to its descriptive scope and we also added a new sentence to clarify the descriptive nature of our results. . There is no claim of causality between Non-CP Cu levels and RGS7, nor there is any kind of speculation which go beyond the current published knowledge about RGS7 and Cu involvement in AD. However, we envisaged potential new or relatively new disease mechanisms to stimulate further research. Some of these concepts have now been stated in the Discussion (page 12, line 327-329, line 411-412 line 420-421, and 436).
2) In figure 1, why was z-scores used instead of absolute values ?
Reply: The data are shown as a z-score, that is, a standardized value of non-Cp Cu showing the variations of the non-Cp Cu values in the subjects’ sample but related to the mean value of non-Cp Cu in controls. The raw data (as numbers) are provided in Table 2. the choice of z-scores is purely cosmetic, as it does not highlight the fact that some non-Cp Cu values are negative. This is, however, evident from the value of the mean and standard deviation provided in Table 2. Furthermore, it is well known that for non-Cp Cu measured by Walshe's calculation, it is possible to obtain negative data. To this particular topic, the authors have devoted extensive discussion in previous articles [12, 19].
3) The results discussed in line 143-144 must be included (at least in supplementary results) as its central to the main findings of the manuscript.
Reply: Data showing that AD patients have higher levels of Cu and Cu:Cp ratio are reported in lines 149-151 and in Figure 1 and in Table 2.
4) Figure 2 should be represented as a beeswarm plot like Figure 1.
Reply: As requested, a new Figure 1 represented as a beeswarm plot like was added in the new revised version of the manuscript
5) Figure 5 is very unclear. It should include details such as (not limiting to): a) species of the sequences compared clearly labeled with identifiers possibly in parenthesis, b) the 19 and/or 14 residues of the extended protein that is conserved should be highlighted clearly (maybe using a box), c) what is the extent of the conservation ? Is it conserved only in mammals or vertebrates or beyond ?
Reply: we structured the reply for a better clarification
5a) Species of sequence are labeled in updated Fig 5.
5b) Extended protein sequence of 19 amino-acids residues is highlighted with a red box and the conserved 14 amino-acids residue sequence from the extended protein sequence is highlighted in the Blue box.
- c) Extended protein sequence is started from 504th position of alignment and ends at 522nd position of alignment (19 amino-acids long) shown in the red box. First five amino-acid residues of extended protein sequence (R (Arginine), K (Lysine), E (Glutamic), G (Glycine), K (Lysine) are not conserved in any sequence but the last 14 amino-acid residues are conserved in all of the sequences except amino-acid residue A (Alanine) at position 519th of alignment which is conserved in Homo Sapiens but replace by V (Valine) in rest of the sequences show in updated fig-5.
- d) it is conserved in vertebrates (Chondrichthyes, reptiles, Aves and mammals).
This information has been added in the Figure Legend of the new Figure 5.
6) Line 246 says "Another result identifies the role of the RGS7 protein in regulating Cu trafficking". This statement is not substantiated as there is no functional link established in relation to RGS7 in the results of this manuscript. If such a role is known, please elaborate. This is akin to my concern mentioned in my first point.
Reply: we rewrote the sentence and tone done the language of this link in the manuscript as suggested; please, see also the reply to comment 1 of rev 2 on the same issue.
Other minor corrections:
Reply: Thanks for nothing those typos
1) line 280 "lowy body" is a typo.
Reply: fixed
2) line 349 "were recruited" repeated twice.
Reply: fixed
3) line 350 "Twenty-8" is a typo.
Reply: fixed
4) A full stop is missing in line 388.
Reply: fixed
5) Figure 3 is very blurry. Please consider uploading a higher resolution figure.
Reply: A new Figure 3 has been provided
Reference for reviewer’s clarification
- Squitti, R.; Ventriglia, M.; Granzotto, A.; Sensi, S. L.; Rongioletti, M. C. A., Non-Ceruloplasmin Copper as a Stratification Biomarker of Alzheimer's Disease Patients: How to Measure and Use It. Curr Alzheimer Res 2021, 18, (7), 533-545.
- Bento, I.; Peixoto, C.; Zaitsev, V. N.; Lindley, P. F., Ceruloplasmin revisited: structural and functional roles of various metal cation-binding sites. Acta Crystallogr D Biol Crystallogr 2007, 63, (Pt 2), 240-8.
- Bielli, P.; Calabrese, L., Structure to function relationships in ceruloplasmin: a 'moonlighting' protein. Cell Mol Life Sci 2002, 59, (9), 1413-27.
- Lopez-Avila, V.; Sharpe, O.; Robinson, W. H., Determination of ceruloplasmin in human serum by SEC-ICPMS. Anal Bioanal Chem 2006, 386, (1), 180-7.
- Twomey, P. J.; Viljoen, A.; House, I. M.; Reynolds, T. M.; Wierzbicki, A. S., Copper:caeruloplasmin ratio. J Clin Pathol 2007, 60, (4), 441-2.
- Squitti, R.; Simonelli, I.; Ventriglia, M.; Siotto, M.; Pasqualetti, P.; Rembach, A.; Doecke, J.; Bush, A. I., Meta-analysis of serum non-ceruloplasmin copper in Alzheimer's disease. J Alzheimers Dis 2014, 38, (4), 809-22.
- Squitti, R.; Ghidoni, R.; Simonelli, I.; Ivanova, I. D.; Colabufo, N. A.; Zuin, M.; Benussi, L.; Binetti, G.; Cassetta, E.; Rongioletti, M.; Siotto, M., Copper dyshomeostasis in Wilson disease and Alzheimer's disease as shown by serum and urine copper indicators. J Trace Elem Med Biol 2018, 45, 181-188.
- Squitti, R.; Siotto, M.; Cassetta, E.; Idrissi, I. G.; Colabufo, N. A., Measurements of serum non-ceruloplasmin copper by a direct fluorescent method specific to Cu(II). Clin Chem Lab Med 2017, 55, (9), 1360 -1367.
Round 2
Reviewer 1 Report
RGS7
There should be an introduction about RGS7 in the introduction section, regarding what it stands for, what is it, what’s the function, why is it relevant to even investigate in the current research. Right now, RGS7 does not show up till in results section, which may confuses the readers.
The authors explained to me why there should be increased level of Aβ-42, so my question remains. I believe in the newer version, it’s in in Line 472, why is there decreased level of Aβ1-42 for AD patients? Is it different from Aβ-42? Shouldn’t there be increased level of Aβ1-42?
Add a citation for benjamini and hochberg’s method of adjustment.
In line 185-188, I don’t the effect sizes for nonsignificant results (after correction for multiple testing) should be reported.
“
- MMSE and MOCA also show similar pattern but not significant, the group difference in these tests may not be significantly different from the group difference in FAB and CDT. ANOVA testing the interaction of patient group (between-subject: CuAD vs AD) and neuropsychological tests (within-subject variable) should be conducted, and only a significant interaction could support that a cognitive deficit in CuAD specific to executive function.
Reply: We referred to a previous reference study (PMID: 15262742) that evaluated if the FBA could contribute to the differential diagnosis of two types of dementia, similar to what we tried to achieve in our pilot study. In that reference study (PMID: 15262742), the authors did not use the repeated measure ANOVA. Furthermore, the cognitive scales we used in the current study have different ranges (FAB has a range of 0-18; MOCA has a range of 0-30; MMSE has a range of 0-30; NPI has a range of 0-14; CDT has a range 0-61) that prevent the application of the repeated measure ANOVA. Thus, we thank the reviewer for the suggestion, but we feel that the repeated measure ANOVA is more suitable for a clinical trial study in which the same index is measured under different conditions so the resulting index's measures are confrontable [e.g. in clinical trial interventions: Sham vs. electrical stimulation or Investigational Medicinal Product (IMP in other words, a new drug) vs. placebo or vs. a Gold standard medication]. However, to accomplish the reviewer’s request for more evidence of the significance of the difference between the two groups, we calculated the ‘d’ Cohen (added also in the Methods, Page 16, lines 507-510). Cohen's d’ can provide a standardized measure of the size of the difference between groups. Cohen’s d demonstrates a large difference between the ‘normal AD’ and the ‘CuAD’ for MMSE (d=0.84, 95%CI: 0.01; 1.68), FBA (d=1.24, 95%CI: 0.37; 2.11), and CDT (d=0.99, 95%CI: 0.15; 1.85), while it is medium for MOCA (d=0.59, 95%CI: -0.23; 1.41) and small for NPI (d=0.26, 95%CI: -1.06; 0.55) (page 5, lines 185-188)”
This did not address my question. The Cohen’s d information does not support the conclusion, let alone 3 of them are not statistically significant after correcting for multiple comparisons. There should be an ANOVA testing the interaction of patient group (between-subject: CuAD vs AD) and neuropsychological tests (within-subject variable) should be conducted, and only a significant interaction could support that a cognitive deficit in CuAD specific to executive function.
In line 184, it should be Cohen’s d….. rather than ‘d’ Cohen
In the response, what does “Out of the label” mean?
I had a hard time understanding the writing of the response... here is an extreme example: an extremely long sentence with lots of commas and special signs, and phrases that I don't understand: "The pilot study proposed, then, is a small-scale application of a diagnostic study devoted to the identification of additional disease-associated metabolic sub-pathways employing an add-on test, to support the identification of causative genetic variants - by means of a WES approach - that can explain the metabolic imbalance observed and that might help to identify therapeutic approach out of the label, based on the genetic variant, to curb the additional disease-associated metabolic imbalance as per the precision medicine principle. "
Author Response
Rev 1 second round replies
Comments and Suggestions for Authors
RGS7
Comment: There should be an introduction about RGS7 in the introduction section, regarding what it stands for, what is it, what’s the function, why is it relevant to even investigate in the current research. Right now, RGS7 does not show up till in results section, which may confuses the readers
Reply: as requested we added some information about RGS7 in the Introduction, and revised the Discussion accordingly (page 3, line 104-122).
Comment: The authors explained to me why there should be increased level of Aβ-42, so my question remains. I believe in the newer version, it’s in in Line 472, why is there decreased level of Aβ1-42 for AD patients? Is it different from Aβ-42? Shouldn’t there be increased level of Aβ1-42?
Reply: Aβ1-42 and Aβ42 are the same. To uniform, we chose to Aβ42 through the manuscript. In the first-round reply, we replied that the changes observed in cerebrospinal fluid include a 50% reduction of amyloid-β 42 as a consequence of amyloid deposition in the brain. In fact, it has been reported that the reduction of Aβ42 in the CSF is a consequence of the fact that it precipitates and forms plaques in the brain. In other words, Aβ42 decrease may reflect plaques acting as an Aβ42 “sink,” hindering the transport of soluble Aβ42 between the brain and CSF (Fagan et al. 2006). This is supported by the evidence of a linear regression analysis that revealed a significant inverse correlation between the overall [11C]PiB uptake in PET and CSF Aβ42 levels (Grimmer et al. 2009). Furthermore, it should be added that while Aβ40 is more produced, Aβ42 is enriched in brain plaque relative to Aβ40 (42 is more hydrophobic and aggregation-prone), so it gets relatively depleted in the soluble pools, such that CSF Aβ42 (and 42/40) ratios go down in AD, i.e. inverse relationship between plaque load and CSF Aβ42/40.
The IGW2 criteria reports: “A marked reduction in CSF Aβ42 and in the Aβ42/Aβ40 ratio” as an established criterion for AD diagnosis per the IGW2 criteria [24], that is the information relevant to our study. We now amended the new version of the manuscript (lines 408-411) and reported these references (added to line 483) and page 15, lines 451-455).
References
Fagan, A. M.; Mintun, M. A.; Mach, R. H.; Lee, S. Y.; Dence, C. S.; Shah, A. R.; LaRossa, G. N.; Spinner, M. L.; Klunk, W. E.; Mathis, C. A.; DeKosky, S. T.; Morris, J. C.; Holtzman, D. M., Inverse relation between in vivo amyloid imaging load and cerebrospinal fluid Abeta42 in humans. Ann Neurol 2006, 59, (3), 512-9.
Grimmer, T.; Riemenschneider, M.; Forstl, H.; Henriksen, G.; Klunk, W. E.; Mathis, C. A.; Shiga, T.; Wester, H. J.; Kurz, A.; Drzezga, A., Beta amyloid in Alzheimer's disease: increased deposition in brain is reflected in reduced concentration in cerebrospinal fluid. Biol Psychiatry 2009, 65, (11), 927-34.
Comment: Add a citation for benjamini and hochberg’s method of adjustment.
Reply:
The reference has been added:
Benjamini Y, Hochberg Y (1995) Controlling the false discovery rate: a practical and powerful approach to multiple hypothesis testing. J R Stat Soc B 57:289–300
Concern: In line 185-188, I don’t the effect sizes for nonsignificant results (after correction for multiple testing) should be reported.
- MMSE and MOCA also show similar pattern but not significant, the group difference in these tests may not be significantly different from the group difference in FAB and CDT. ANOVA testing the interaction of patient group (between-subject: CuAD vs AD) and neuropsychological tests (within-subject variable) should be conducted, and only a significant interaction could support that a cognitive deficit in CuAD specific to executive function.
Reply: We referred to a previous reference study (PMID: 15262742) that evaluated if the FBA could contribute to the differential diagnosis of two types of dementia, similar to what we tried to achieve in our pilot study. In that reference study (PMID: 15262742), the authors did not use the repeated measure ANOVA. Furthermore, the cognitive scales we used in the current study have different ranges (FAB has a range of 0-18; MOCA has a range of 0-30; MMSE has a range of 0-30; NPI has a range of 0-14; CDT has a range 0-61) that prevent the application of the repeated measure ANOVA. Thus, we thank the reviewer for the suggestion, but we feel that the repeated measure ANOVA is more suitable for a clinical trial study in which the same index is measured under different conditions so the resulting index's measures are confrontable [e.g. in clinical trial interventions: Sham vs. electrical stimulation or Investigational Medicinal Product (IMP in other words, a new drug) vs. placebo or vs. a Gold standard medication]. However, to accomplish the reviewer’s request for more evidence of the significance of the difference between the two groups, we calculated the ‘d’ Cohen (added also in the Methods, Page 16, lines 507-510). Cohen's d’ can provide a standardized measure of the size of the difference between groups. Cohen’s d demonstrates a large difference between the ‘normal AD’ and the ‘CuAD’ for MMSE (d=0.84, 95%CI: 0.01; 1.68), FBA (d=1.24, 95%CI: 0.37; 2.11), and CDT (d=0.99, 95%CI: 0.15; 1.85), while it is medium for MOCA (d=0.59, 95%CI: -0.23; 1.41) and small for NPI (d=0.26, 95%CI: -1.06; 0.55) (page 5, lines 185-188)”
This did not address my question. The Cohen’s d information does not support the conclusion, let alone 3 of them are not statistically significant after correcting for multiple comparisons. There should be an ANOVA testing the interaction of patient group (between-subject: CuAD vs AD) and neuropsychological tests (within-subject variable) should be conducted, and only a significant interaction could support that a cognitive deficit in CuAD specific to executive function.
Reply: We run an ANOVA study as suggested. To perform a repeated measures ANOVA considering the patient group as a between-subject factor and all the neuropsychological tests as a within-subject factor, each test was normalized (based on the formula: ((observed value-Scale Minimum value))⁄( Scale Maximum value-Scale Minimum value). The model showed a significant interaction Type of Test x Patient group effect (p=0.021). The post hoc comparisons showed significantly higher values in AD compared to CuAD in FAB (B-H adjusted p=0.006) as well as in CDT (B-H adjusted p=0.024), borderline significant in MMSE (B-H adjusted p=0.055). We reported in the text the Benjamini & Hochberg [43] p-value adjustment for multiple testing (B-H adjusted).
We thank the reviews for this suggestion that provide additional confirmation of the robustness of our data.
Comment: In line 184, it should be Cohen’s d….. rather than ‘d’ Cohen
Reply: done
Comment: In the response, what does “Out of the label” mean?
Reply: the reviewer is correct; the exact English term was ‘Off-label’ Drug!
An off-label drug is a drug approved for treating one type of disease but is used to treat a different disease: this use is an off-label use. The same concept of drug repurposing. In our case, we were talking of use off-label Zinc that has been approved to treat Wilson's disease, and we would like to use it to treat Alzheimer’s disease
I had a hard time understanding the writing of the response... here is an extreme example: an extremely long sentence with lots of commas and special signs, and phrases that I don't understand: "The pilot study proposed, then, is a small-scale application of a diagnostic study devoted to the identification of additional disease-associated metabolic sub-pathways employing an add-on test, to support the identification of causative genetic variants - by means of a WES approach - that can explain the metabolic imbalance observed and that might help to identify therapeutic approach out of the label, based on the genetic variant, to curb the additional disease-associated metabolic imbalance as per the precision medicine principle. "
Reply: Sorry, for the long replies to the first-round review. it was tentative to explain all the concepts since we are aware that the topic is very specific and is based on previous evidence. The intention was to provide the reviewer with the best reply, but we failed. We did an English editing on the revised version of the manuscript. In this second-round review, replies have been kept short.
The paragraphs reported above were only present in the reply to the reviewer but not in the manuscript. In the text of the manuscript the same concept is reported in short on page 15, lines 455-475).
